# Joint Entropy Search
# for Maximally-Informed Bayesian Optimization

**Carl Hvarfner**
carl.hvarfner@cs.lth.se
Lund University

**Frank Hutter**
fh@cs.uni-freiburg.de
University of Freiburg
Bosch Center for Artificial Intelligence

**Luigi Nardi**
luigi.nardi@cs.lth.se
Lund University
Stanford University
DBtune

## Abstract

Information-theoretic Bayesian optimization techniques have become popular for optimizing expensive-to-evaluate black-box functions due to their non-myopic qualities. Entropy Search and Predictive Entropy Search both consider the entropy over the optimum in the input space, while the recent Max-value Entropy Search considers the entropy over the optimal value in the output space. We propose Joint Entropy Search (JES), a novel information-theoretic acquisition function that considers an entirely new quantity, namely the entropy over the joint optimal probability density over both input and output space. To incorporate this information, we consider the reduction in entropy from conditioning on fantasized optimal input/output pairs. The resulting approach primarily relies on standard GP machinery and removes complex approximations typically associated with information-theoretic methods. With minimal computational overhead, JES shows superior decision-making, and yields state-of-the-art performance for information-theoretic approaches across a wide suite of tasks. As a light-weight approach with superior results, JES provides a new go-to acquisition function for Bayesian optimization.

## 1 Introduction

The optimization of expensive black-box functions is a prominent task, arising across a wide range of applications. *Bayesian optimization* (BO) [25, 35] is a sample-efficient approach, and has been successfully applied to various problems, including machine learning hyperparameter optimization [2, 20, 33, 37], robotics [3, 6, 23, 24], hardware design [11, 27], and tuning reinforcement learning agents like AlphaGo [7]. In BO, a probabilistic surrogate model is used for modeling the (unknown) objective. The selection policy employed by the BO algorithm is dictated by an acquisition function, which draws on the uncertainty of the surrogate to guide the selection of the next query.

The choice of acquisition function is significant for the success of the BO algorithm. A popular line of acquisition functions takes an information-theoretic angle, and considers the *expected information gain* regarding the location of the optimum that is obtained from an upcoming query. *Entropy Search* (ES) [15], *Predictive Entropy Search* (PES) [16] and the earlier work of IAGO [46] select queries by maximizing this quantity. While ES and PES are efficient in the number of queries to optimize the objective, they both require significant computational effort and complex approximations of the expected information gain, which impacts their performance and practical use [16, 47].

36th Conference on Neural Information Processing Systems (NeurIPS 2022).

A related information-theoretic family of approaches considers the information gain on the optimal objective value [18, 31, 47]. *Max-value Entropy Search* (MES) [47] was the first information-theoretic approach to have a proven convergence rate, albeit only in a noiseless problem setting. Moreover, its consideration of a one-dimensional density over the output space as opposed to a $D$-dimensional input space and a reduction in intricate approximations yielded a computationally efficient alternative to the ES/PES line of approaches. Despite its empirical success, some crucial shortcomings of MES have been highlighted in recent works. Its convergence rate has been disputed [42], and crucially, it does not differentiate between the (unobserved) maximal objective value $f^*$ and the observed noisy maximum $y_{max}$ [26, 28, 41, 42]. As such, its assumption on the posterior distribution of the output $p(y|\mathcal{D}, \boldsymbol{x})$ does not hold in any setting where noise is present, and several follow-ups have been proposed to address the noisy problem setting [26, 28, 41, 42].

We propose an approach which merges the ES/PES and MES lines of work, and provides an all-encompassing perspective on information gain regarding optimality. We introduce Joint Entropy Search (JES), a novel acquisition function which has the following advantages over existing infomation-theoretic approaches:

1. It utilizes two sources of information, by considering the entropy over both the optimum and the noiseless optimal value;

2. It utilizes the full optimal observation, allowing it to rely primarily on exact computation through standard GP machinery instead of complex approximations; and

3. It is computationally light-weight, requiring minimal pre-computation relative to other information-theoretic approaches which consider the input space.

Simultaneously to our work, a similar approach aimed at the multi-objective setting, was proposed by Tu et al. [44]. The authors independently came up with the same JES acquisition function, with a subtly different approximation scheme to the one we present. We see our work as being complementary to theirs because we both demonstrate the effectiveness of this new acquisition function in different settings - theirs being multi-objective and batch evaluations, ours being single-objective and large levels of output noise. Our code for reproducing the experiments is available at `https://github.com/hvarfner/JointEntropySearch`.

## 2 Background and related work

**Bayesian optimization.**    We consider the problem of optimizing a black-box function $f$ across a set of feasible inputs $\mathcal{X} \subset \mathbb{R}^d$:

$$\boldsymbol{x}^* \in \arg\max_{\boldsymbol{x} \in \mathcal{X}} f(\boldsymbol{x}). \tag{1}$$

We assume that $f(\boldsymbol{x})$ is expensive to evaluate, and can potentially only be observed through a noise-corrupted estimate, $y$, where $y = f(\boldsymbol{x}) + \epsilon, \epsilon \sim \mathcal{N}(0, \sigma_\epsilon^2)$ for some noise level $\sigma_\epsilon^2$. In this setting, we wish to maximize $f$ in an efficient manner, typically while adhering to a budget which sets a cap on the number of points that can be evaluated. BO aims to globally maximize $f$ by an initial design and thereafter sequentially choosing new points $\boldsymbol{x}_n$ for some iteration $n$, creating the data $\mathcal{D}_n = \mathcal{D}_{n-1} \cup \{(\boldsymbol{x}_n, y_n)\}$. After each new observation, BO constructs a probabilistic surrogate model $p(f|\mathcal{D}_n)$ and uses that surrogate to build an acquisition function $\alpha(\boldsymbol{x}, \mathcal{D}_n)$. The combination of surrogate model and acquisition function encodes the strategy for selecting the next point $\boldsymbol{x}_{n+1}$. After the full budget of $N$ iterations is exhausted, a best configuration $\boldsymbol{x}_N^*$ is returned as either the $\arg\max$ of the observed values, or the optimum as predicted by the surrogate model.

**Gaussian processes.**    When constructing the surrogate, the most common choice is *Gaussian processes* (GPs) [30]. Formally, a GP is an infinite collection of random variables, such that every finite subset of those variables follows a multivariate Gaussian distribution. The GP utilizes a covariance function $k$, which encodes a prior belief for the smoothness of $f$, and determines how previous observations influence prediction. Given observations $\mathcal{D}_n$ at iteration $n$, the posterior $p(f|\mathcal{D}_n)$ over the objective is characterized by the posterior mean $m_n$ and variance $s_n$ of the GP:

$$m_n(\boldsymbol{x}) = \mathbf{k}_n(\boldsymbol{x})^\top (\mathbf{K}_n + \sigma_\epsilon^2 \mathbf{I})^{-1} \mathbf{y}, \quad s_n(\boldsymbol{x}) = k(\boldsymbol{x}, \boldsymbol{x}) - \mathbf{k}_n(\boldsymbol{x})^\top (\mathbf{K}_n + \sigma_\epsilon^2 \mathbf{I})^{-1} \mathbf{k}_n(\boldsymbol{x}), \tag{2}$$

where $(\mathbf{K}_n)_{ij} = k(\boldsymbol{x}_i, \boldsymbol{x}_j)$, $\mathbf{k}_n(\boldsymbol{x}) = [k(\boldsymbol{x}, \boldsymbol{x}_1), \ldots, k(\boldsymbol{x}, \boldsymbol{x}_n)]^\top$ and $\sigma_\epsilon^2$ is the noise variance. Alternative surrogate models include random forests [19] and Bayesian neural networks [38, 39].

**Acquisition functions.** The *acquisition function* acts on the surrogate model to quantify the attractiveness of a point in the search space. Acquisition functions employ a trade-off between exploration and exploitation, typically using a greedy heuristic to do so. Simple, computationally cheap heuristics are Expected Improvement (`EI`) [5, 21]. For a noiseless function, `EI` selects the next point $\boldsymbol{x}_{n+1}$ as

$$\boldsymbol{x}_{n+1} \in \arg\max_{\boldsymbol{x}\in\mathcal{X}} \mathbb{E}\left[(y_n^* - y_{n+1}^*)^+\right] = \arg\max_{\boldsymbol{x}\in\mathcal{X}} Z s_n(\boldsymbol{x})\Phi(Z) + s_n(\boldsymbol{x})\phi(Z), \quad (3)$$

where $Z = (y_n^* - m_n(\boldsymbol{x}))/s_n(\boldsymbol{x})$. Other acquisition functions which use similar heuristics are the Upper Confidence Bound (`UCB`) [40], and Probability of Improvement (`PI`) [22]. A more sophisticated approach related to `EI` is Knowledge Gradient (`KG`) [12].

**Information-theoretic acquisition functions.** Information-theoretic acquisition functions [15, 16, 32, 47] and their various adaptations [1, 17, 34] seek to maximize the expected information gain $I$ from observing a subsequent query $(\boldsymbol{x}, y)$ regarding the optimum, $\boldsymbol{x}^*$. This equates to reducing the uncertainty of the density over the optimum, $p(\boldsymbol{x}^*|\mathcal{D}) = \mathbb{P}(\boldsymbol{x} = \arg\max_{\boldsymbol{x}'\in\mathcal{X}} f(\boldsymbol{x}')|\mathcal{D})$, using the information obtained through $(\boldsymbol{x}, y)$. By quantifying uncertainty through the differential entropy H, design points are selected based on the expected reduction in this quantity over $p(\boldsymbol{x}^*|\mathcal{D})$. Formally, this is expressed as the difference between the current entropy over $p(\boldsymbol{x}^*|\mathcal{D})$, and the expected entropy of that density after observing the next query:

$$\alpha_{\text{ES}}(\boldsymbol{x}) = I((\boldsymbol{x}, y); \boldsymbol{x}^*|\mathcal{D}) = \text{H}[p(\boldsymbol{x}^*|\mathcal{D})] - \mathbb{E}_y\left[\text{H}[p(\boldsymbol{x}^*|\mathcal{D}\cup(\boldsymbol{x}, y))]\right]. \quad (4)$$

By utilizing the symmetric property of the mutual information, one can arrive at an equivalent expression, where the entropy is computed with regard to the density over the output $y$,

$$\alpha_{\text{PES}}(\boldsymbol{x}) = I(y; (\boldsymbol{x}, \boldsymbol{x}^*)|\mathcal{D}) = \text{H}[p(y|\mathcal{D}, \boldsymbol{x})] - \mathbb{E}_{\boldsymbol{x}^*}\left[\text{H}[p(y|\mathcal{D}, \boldsymbol{x}, \boldsymbol{x}^*)]\right]. \quad (5)$$

Eq. 4 is the original formulation used in ES [15] and Eq. 5 is the formulation introduced with PES [16]. Both formulations require a series of approximations and expensive computational steps to compute the entropy in the second term. For PES specifically, with $n$ data points of dimension $d$, the second term is estimated through Monte Carlo (MC) methods by computing Cholesky decompositions of size $\mathcal{O}(n + d^2/2)^3$, and approximating the Hessian at the optimum for each MC sample.

MES [47] avoids this computational hurdle by considering the information gain $I((\boldsymbol{x}, y); y^*|\mathcal{D})$ regarding the optimal value $y^*$. As such, it computes the entropy reduction for a one-dimensional density:

$$\alpha_{\text{MES}}(\boldsymbol{x}) = I(y; (\boldsymbol{x}, y^*)|\mathcal{D}) = \text{H}[p(y|\mathcal{D}, \boldsymbol{x})] - \mathbb{E}_{y^*}\left[\text{H}[p(y|\mathcal{D}, \boldsymbol{x}, y^*)]\right]. \quad (6)$$

Here, it is assumed that the posterior predictive distribution $p(y|\mathcal{D}, \boldsymbol{x}, y^*)$ is a truncated Gaussian distribution, for which the entropy can be computed in closed form. However, $p(y|\mathcal{D}, \boldsymbol{x}, y^*)$ takes this form only in a strictly noiseless setting [28, 41], where it holds true that $f^* = y_{max}$, i.e. when the maximal observation and the optimal value of the objective function coincide. For noisy applications, this assumption leads to an overestimation of the entropy reduction [28].

## 3 Joint Entropy Search

We now present Joint Entropy Search (`JES`), a novel information-theoretic approach for Bayesian optimization. As for other information-theoretic acquisition functions, `JES` considers a mutual information quantity. However, unlike its predecessors, `JES` adds an additional piece of information: compared to ES/PES, it adds the density over the noiseless optimal value $f^*$, and compared to MES, it adds the density over $\boldsymbol{x}^*$. It utilizes a novel two-step reduction in the predictive entropy from conditioning on sampled optima and their associated values. Throughout the section, we will refer to a sampled optimum and its associated value, $(\boldsymbol{x}^*, f^*)$, as an *optimal pair*.

### 3.1 Joint density over the optimum and optimal value: pictorial

JES considers the joint probability density $p(\boldsymbol{x}^*, f^*)$ over both the optimum $\boldsymbol{x}^*$ and the true, noiseless optimal value $f^*$. Fig. 1 visualizes the densities $p(\boldsymbol{x}^*)$ and $p(f^*)$, considered by ES/PES and MES, respectively, and the joint density $p(\boldsymbol{x}^*, f^*)$, considered by JES. As highlighted by the vertical dashed lines for the point selection of each strategy (bottom), PES chooses strictly to reduce the

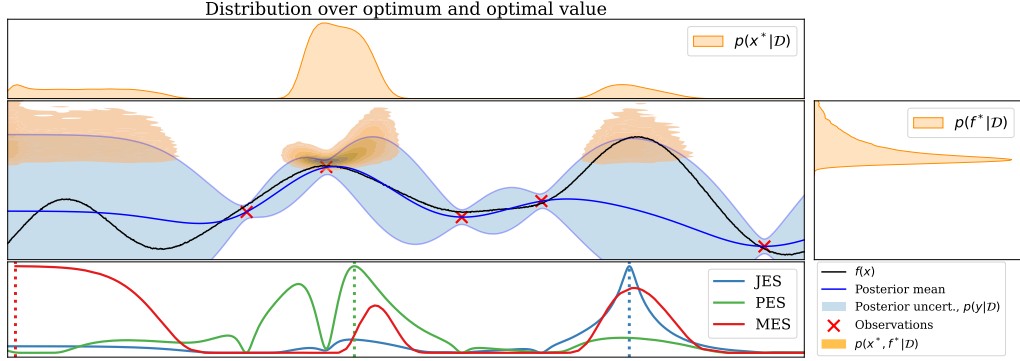

**Figure 1:** The densities considered by ES/PES (top), MES (right) and JES (center) on a one-dimensional toy example. The multimodal density $p(\boldsymbol{x}^*, f^*)$ is reduced to a heavy-tailed density over $f^*$ for the density used by MES (right), which does not capture the multi-modality of the density over the optimum. The density over $\boldsymbol{x}^*$ used by PES (top) does not capture the apparent exploration/exploitation trade-off that exists between the modes. The acquisition functions and their next point selections are highlighted with dashed lines (bottom).

uncertainty over $\boldsymbol{x}^*$, and as such, considers a region where the uncertainty over the optimal value is low. However, it can effectively determine that the right side of the local optimum is more promising to query next. MES seeks to reduce the tail of the probability density over $f^*$ (right), which in this case leads to an exploratory query. JES' joint probability density over optimum and optimal value captures uncertainties over both "where" and "how large" the optimum will be. As such, it selects a point which is uncertain under both measures. As such, JES will learn about likely locations for the optimum, while simultaneously learning probable lower and upper bounds for the optimal value, which by itself yields an effective query strategy [47] and provides valuable knowledge for future queries. For the selected query in Fig. 1, JES will learn substantially about both $\boldsymbol{x}^*$ and $f^*$ by querying it, whereas PES and MES learn only about one of them.

### 3.2    The Joint Entropy Search acquisition function

We consider the mutual information between the random variables $(\boldsymbol{x}^*, f^*)$ and a future query $(\boldsymbol{x}, y)$:

$$\alpha_{\mathrm{JES}}(\boldsymbol{x}) = I((\boldsymbol{x}, y); (\boldsymbol{x}^*, f^*)|\mathcal{D}_n) \tag{7}$$

$$= \mathrm{H}[p(y|\mathcal{D}, \boldsymbol{x})] - \mathbb{E}_{(\boldsymbol{x}^*, f^*)}\left[\mathrm{H}[p(y|\mathcal{D}, \boldsymbol{x}, \boldsymbol{x}^*, f^*)]\right] \tag{8}$$

$$= \mathrm{H}[p(y|\mathcal{D}, \boldsymbol{x})] - \mathbb{E}_{(\boldsymbol{x}^*, f^*)}\left[\mathrm{H}[p(y|\mathcal{D} \cup (\boldsymbol{x}^*, f^*), \boldsymbol{x}, f^*)]\right]. \tag{9}$$

Eq. 8 is similar to Eq. 6 but with the addition of $\boldsymbol{x}^*$ and the replacement of $y^*$ with $f^*$ in the conditioning of the second term. The expectation is computed with respect to a $D + 1$-dimensional joint probability density over $\boldsymbol{x}^*$ and $f^*$. In Eq. 9, we make it explicit that the conditional density inside the expectation is obtained after 1. conditioning the GP on the previous data $\mathcal{D}$, plus one additional noiseless optimal pair $(\boldsymbol{x}^*, f^*)$, and 2. knowing that the noiseless optimal value is in fact $f^*$. By utilizing the complete observation $(\boldsymbol{x}^*, f^*)$, we can treat it like any (noiseless) observation. As such, we quantify much of the entropy reduction by utilizing standard GP conditioning functionality. For 2., we cannot globally condition on $f(\boldsymbol{x}') \leq f^*, \forall \boldsymbol{x}'$. As such, we follow previous work [26, 28, 42, 47] and enforce the condition *locally* at the current query $\boldsymbol{x}$. The resulting effect is to truncate the GP's posterior over $f$ locally at $\boldsymbol{x}$, upper bounding it to $f^*$. Notably, utilizing the fantasized observation $(\boldsymbol{x}^*, f^*)$ guarantees that the conditioned optimal value $f^*$ in JES is actually obtained, rather than serving as a possibly unattained upper bound, which is typical in the MES family of acquisition functions. The expectation in Eq. 9 is approximated through MC by sampling $L$ optimal pairs $\{(\boldsymbol{x}_\ell^*, f_\ell^*)\}_{\ell=1}^L$ from $p(\boldsymbol{x}^*, f^*)$ using an approximate version of *Thompson Sampling* (TS) [43], as explained in Sec. 3.3. In Fig. 2, the resulting posterior distribution of the two-step conditioning is shown in greater detail. As pointed out in [28, 41], after conditioning on $f^*$, the posterior predictive density over $y$ is a sum of a truncated Gaussian distribution over $f$ and the Gaussian noise $\epsilon$. The entropy reduction from the two-step conditioning yields two separate variance reduction steps over $p(y|\mathcal{D}, \boldsymbol{x})$: a conditioning term and a truncation term. The former is computed exactly, while the latter, generally smaller term, requires approximation, as shown in Sec. 3.4.

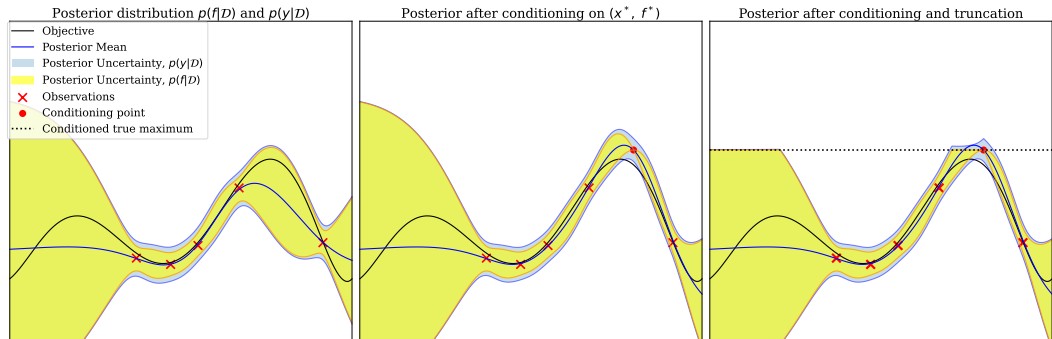

**Figure 2:** Step-by-step modeling when conditioning on one optimal pair $(\boldsymbol{x}^*, f^*)$. The posterior with noise $p(y|\mathcal{D})$ and without noise $p(f|\mathcal{D})$ are illustrated in blue and yellow, respectively. The GP after 5 (noisy) observations, before conditioning on $(\boldsymbol{x}^*, f^*)$ is shown on the left. In the middle panel, we draw $(\boldsymbol{x}^*, f^*)$ and condition on it, making $p(f|\mathcal{D} \cup (\boldsymbol{x}^*, f^*))$ a delta distribution at the conditioning point as the fantasized observation $f^*$ is noiseless. Since $f^*$ is also the presumed noiseless maximum, we truncate its posterior $p(f|\mathcal{D} \cup (\boldsymbol{x}^*, f^*), f^*)$ globally in the right panel. The observation noise allows for non-zero density on $p(y > f^*|\mathcal{D} \cup (\boldsymbol{x}^*, f^*), f^*)$. We note that, while the noise is homoscedastic, its relative contribution to the total variance differs over the input space. As such, and since we're plotting standard deviations (not variances), the blue region is wider near observed data, where $p(f|\mathcal{D})$ has lower variance.

Fig. 3 shows the difference in log variance over $p(y|\mathcal{D}, \boldsymbol{x})$ resulting from conditioning (in blue) and truncation (in orange) for the scenario in Fig. 2. The overall reduction is largest close to the point of conditioning, and the truncation term mainly contributes at uncertain regions far away from the conditioned point. Moreover, the magnitude of the conditioning term will rely on the prior variance at the conditioned point, as a larger prior variance will lead to a larger reduction in entropy from conditioning. As we average over optimal pairs, many such entropy reduction terms accumulate.

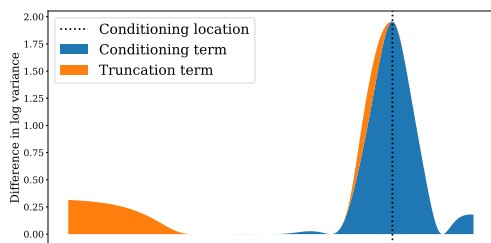

**Figure 3:** Reduction in log variance from the conditioning step and the truncation step as visualized in Fig. 2. The local conditioning term (blue), and the globally variance-reducing truncation term (orange).

### 3.3 Incorporating optimal pairs

To obtain samples $(\boldsymbol{x}^*, f^*)$, we utilize an approximate variant of TS [43], originally proposed in PES [16]. We utilize Bochner's theorem [4], which, for any stationary kernel $k$, asserts the existence of its Fourier dual $s(\boldsymbol{w})$. By normalizing $s(\boldsymbol{w})$, we obtain the spectral density $p(\boldsymbol{w}) = s(\boldsymbol{w})/\alpha$, where $\alpha$ is a normalization constant. We can then write the kernel as an expectation,

$$k(\boldsymbol{x}, \boldsymbol{x}') = \alpha\mathbb{E}_{\boldsymbol{w}}[e^{i\boldsymbol{w}^\intercal(\boldsymbol{x}-\boldsymbol{x}')}] = 2\alpha\mathbb{E}_{\boldsymbol{w},\boldsymbol{b}}[\cos(\boldsymbol{w}^\intercal\boldsymbol{x} + \boldsymbol{b})\cos(\boldsymbol{w}^\intercal\boldsymbol{x}' + \boldsymbol{b})], \tag{10}$$

where $\boldsymbol{b} \sim \mathcal{U}(\boldsymbol{0}, 2\pi\boldsymbol{I})$. Following Rahimi and Recht [29], we sample $\boldsymbol{b}$ and $\boldsymbol{w}$ to obtain an unbiased estimate of the kernel $k$. From this approximation, approximate sample paths can be drawn as a weighted sum of basis functions. This form allows for fast and dense querying of the sample paths – the $\arg\max$ and $\max$ of which is an approximate draw from $p(\boldsymbol{x}^*, f^*)$. In PES, each sample $\boldsymbol{x}_\ell^*$ along with its inverted Hessian is required for computing the acquisition function. To obtain the Hessian, each sample needs to be thoroughly optimized through gradient-based optimization. JES on the other hand, only requires $(\boldsymbol{x}^*, f^*)$. As such, it can rely on cheap, approximate optimization of these samples, e.g., by densely querying sample points on a non-uniform grid.

After obtaining a set of optimal pairs $\{(\boldsymbol{x}_\ell^*, y_\ell^*)\}_{\ell=1}^L$, JES computes the conditional entropy quantity over the output $y$. Concretely, we generate $L$ GPs, each modeling a posterior density $\{p(y|\mathcal{D} \cup (\boldsymbol{x}_\ell^*, f_\ell^*), \boldsymbol{x})\}_{\ell=1}^L$ conditioned on an optimal pair and previously observed data $\mathcal{D}$. Since each optimal pair is drawn from the current GP hyperparameter set, we know that the current hyperparameter set is the correct one even after adding the optimal pair to the data. By consequence, JES can compute the updated inverse Gram matrix, $(K + \sigma_\epsilon^2)^{-1}$, through a rank-1 update, instead of solving a linear system

of equations. Utilizing the Sherman–Morrison formula [36], we obtain updated Gram matrices in $\mathcal{O}(n^2)$ for each sample, as opposed to $\mathcal{O}(n^3)$ for solving the linear system of equations.

### 3.4 Approximating the truncated entropy

As highlighted in the right panel of Fig. 2, conditioning on $f^*$ yields a truncated normal distribution $p(f|\mathcal{D} \cup (\boldsymbol{x}^*, f^*), \boldsymbol{x}, f^*)$ after having locally enforced the inequality $f(\boldsymbol{x}) \leq f^*$. The entropy, however, is computed with regard to the density over noisy observations, $y = f + \epsilon$, which follows an Extended Skew distribution [28] and as such, does not have tractable entropy. We approximate this quantity through moment matching [26] of the truncated Gaussian distribution over $f$, which yields a valid lower bound on the information gain [26]. Consequently, we obtain two Gaussian densities $\hat{p}(f|\mathcal{D} \cup (\boldsymbol{x}^*, f^*), \boldsymbol{x}, f^*) \sim \mathcal{N}(m_{f|f^*}, \sigma^2_{f|f^*})$ and $p(\epsilon) \sim \mathcal{N}(0, \sigma^2_\epsilon)$, where $m_{f|f^*}$ and $\sigma^2_{f|f^*}$ are the mean and variance of the truncated Gaussian posterior $p(f|\mathcal{D} \cup (\boldsymbol{x}^*, f^*), \boldsymbol{x}, f^*)$. Due to independence between $f$ and $\epsilon$ and the linearity of Gaussian distributions, we can then compute the entropy of the approximate density $\hat{p}_y$ exactly as $\text{H}[\hat{p}(y|\mathcal{D} \cup (\boldsymbol{x}^*, f^*), \boldsymbol{x}, f^*)] = \log(2\pi(\sigma^2_\epsilon + \sigma^2_{f|f^*}))$. Moreover, the variance of the truncated Gaussian $\sigma^2_{f|f^*}$ is computed as

$$\sigma^2_{f|f^*}(\boldsymbol{x}; \mathcal{D} \cup (\boldsymbol{x}^*_\ell, f^*_\ell)) = \sigma^2_T(f^*; m^\ell_n(\boldsymbol{x}), s^\ell_n(\boldsymbol{x})) \tag{11}$$

where $\sigma^2_T(\alpha; \mu, \sigma^2)$ is the variance of an upper truncated Gaussian distribution with parameters $(\mu, \sigma^2)$, truncated at $\alpha$, and $m^\ell_n(\boldsymbol{x})$ and $s^\ell_n(\boldsymbol{x})$ are the mean and covariance functions of the GP which is conditioned on the optimal pair $(\boldsymbol{x}^*_\ell, f^*_\ell)$. The quality of the moment matching approximation is studied in greater detail in Appendix E.

### 3.5 Exploitative selection to guard against model misspecification

As with all information-theoretic approaches, JES aims to reduce the uncertainty over the location of the optimum. With this strategy, the incentive to query the perceived optimum is often lower than for heuristic approaches, such as EI. In cases where the surrogate model is misspecified, information-theoretic approaches risk reducing the entropy based on a faulty belief of the optimum, which can drastically impact their performance. As a remedy, we utilize a $\gamma$-exploit approach inspired by the parallel context of AEGIS [9]: with probability $\gamma$, JES will query the $\arg\max$ of the posterior mean to confirm its belief of the location of the optimum. If the model is misspecified, these exploitative steps enable the algorithm to reconsider its beliefs, rather than continuing to act based on faulty ones. In Appendix C.2, we show how this approach can substantially improve performance in cases of surrogate model misspecification, while having negligible impact on performance in the worst case.

### 3.6 Putting it all together: The JES algorithm

For a sampled set of size $L$, containing optimal pairs $\{(\boldsymbol{x}^*_\ell, y^*_\ell)\}^L_{\ell=1}$ and GPs with mean and covariance functions $\{m^\ell_n(\boldsymbol{x}), s^\ell_n(\boldsymbol{x})\}^L_{\ell=1}$, the expression for the JES acquisition function is

$$\alpha(\boldsymbol{x})_{\text{JES}} = \text{H}[p(y|\mathcal{D}, \boldsymbol{x})] - \mathbb{E}_{(\boldsymbol{x}^*, f^*)}\left[\text{H}[p(y|\mathcal{D} \cup (\boldsymbol{x}^*, f^*), \boldsymbol{x}, f^*)]\right] \tag{12}$$

$$\approx \log(2\pi(s_n(\boldsymbol{x}) + \sigma^2_\epsilon)) - \frac{1}{L}\sum_{\ell=1}^{L}\log(2\pi(\sigma^2_\epsilon + \sigma^2_{f|f^*}(\boldsymbol{x}; \mathcal{D} \cup (\boldsymbol{x}^*_\ell, f^*_\ell)))), \tag{13}$$

The first term in 13 is simply the entropy of a Gaussian that can be computed in closed form. The second term contains both the conditioning term, which is exact, and the truncation, which is approximated as described in Sec. 3.4. Algorithm 1 outlines pseudocode for JES in its entirety.

## 4 Experimental evaluation

**Benchmarks.** We now evaluate JES on a suite of diverse tasks. We consider three different types of benchmarks: samples drawn from a GP prior, commonly used synthetic test functions [16], and a collection of classification tasks on tabular data using an MLP, provided through HPOBench [10]. For the GP prior tasks, the hyperparameters are known for all methods to evaluate the effect of the acquisition function in isolation. Consequently, we do not use the $\gamma$-exploit approach from Sec. 3.5

**Algorithm 1** JES Algorithm
___
1: **Input:** Black-box function $f$, input space $\mathcal{X}$, size $M$ of the initial design, max number of optimization iterations $N$, number of posterior MC samples $L$, fraction of exploit samples $\gamma$.
2: **Output:** Optimized design $\boldsymbol{x}^*$.
3: $\mathcal{D}_M \leftarrow \{(\boldsymbol{x}_i, y_i)\}_{i=1}^M$              ▷ initial design
4: **for** $\{n = M+1, \ldots, M+N\}$ **do**
5:   $m(\boldsymbol{x}), s^2(\boldsymbol{x}) \leftarrow \text{FITGP}(\mathcal{D}_{n-1})$
6:   **if** $\text{RAND}(0,1) < \gamma$ **then**   $\boldsymbol{x}_n \leftarrow \arg\max_{\boldsymbol{x} \in \mathcal{X}} m_n(\boldsymbol{x})$    ▷ as described in Sec. 3.5
7:   **else**
8:    **for** $\{\ell = 1, \ldots, L\}$ **do**
9:     $(\boldsymbol{x}_\ell^*, y_\ell^*) \leftarrow \text{TS}(f)$           ▷ as described in Sec. 3.3
10:     $p(y|\mathcal{D}_{n-1} \cup (\boldsymbol{x}_\ell^*, f_\ell^*), \boldsymbol{x}) \leftarrow \text{UPDATEGP}(\boldsymbol{x}_\ell^*, f_\ell^*)$   ▷ as described in Sec. 3.3
11:    **end for**
12:    $\boldsymbol{x}_n = \arg\max_{\mathcal{X}} \alpha_{\text{JES}}(\boldsymbol{x})$        ▷ defined in Eq. 12
13:   **end if**
14:   $y_n = f(\boldsymbol{x}_n) + \epsilon, \quad \mathcal{D}_n \leftarrow \mathcal{D}_{n-1} \cup \{(\boldsymbol{x}_n, y_n)\}$    ▷ observe next query
15: **end for**
16: **return** $\boldsymbol{x}^* \leftarrow \arg\max_{\boldsymbol{x} \in \mathcal{X}} m_n(\boldsymbol{x})$
___

in this case (i.e., we set $\gamma = 0$ in Algorithm 1). For the synthetic and MLP tasks, we marginalize over the GP hyperparameters, and set $\gamma = 0.1$. The hyperparameters of the GP prior experiments can be found in Appendix B, and ablation studies on $\gamma$ in Appendix C.

**Evaluation criteria.** We use two types of evaluation criteria as in [47]: *simple regret* and *inference regret*. The simple regret $r_n = \max_{\boldsymbol{x} \in \mathcal{X}} f(\boldsymbol{x}) - \max_{t \in [1,n]} f(\boldsymbol{x}_t)$ measures the value of the best queried point so far. After a query, we may infer an $\arg\max$ of the function, which is chosen as $\boldsymbol{x}_n^* = \arg\max_{\boldsymbol{x} \in \mathcal{X}} m_n(\boldsymbol{x})$ [15, 16, 47]. We denote the inference regret as $r_n = \max_{\boldsymbol{x} \in \mathcal{X}} f(\boldsymbol{x}) - f(\boldsymbol{x}_n^*)$. Since information-theoretic approaches do not necessarily seek to query the optimum, but only to know its location, inference regret characterizes how satisfying our belief of the $\arg\max$ is. Notably, this metric is non-monotonic, meaning that the best guess can worsen with time. We use this metric in the ideal model benchmarking setting, when we sample tasks from a GP with known hyperparameters. We use simple regret for the synthetic test functions, as it constitutes a metric that is more robust to surrogate model misspecification. Inference regret for these tasks can be found in Appendix F. For the HPOBench tasks, inference regret is unobtainable.

**The experimental setup.** We compare against other state-of-the-art information-theoretic approaches: PES [16] and MES [47], as well as EI [21]. The acquisition functions are all run in the same framework written in MATLAB, created for the original PES implementation by Hernández-Lobato et al. [16]. All synthetic experiments were run for $50D$ iterations. In the main paper, we fix the number of MC samples for MES, PES and JES to 100 each. In Appendix D we assess the sensitivity of JES to this number and quantify the computational expense. In Appendix B, we provide all details on our experimental setup, including the runtime tests.

### 4.1 GP prior samples

We consider samples from a GP prior for four different dimensionalities: 2D, 4D, 6D, and 12D, with a noise standard deviation of $0.1$ for a range of outputs spanning roughly $[-10, 10]$. These tasks constitute an optimal setting for each algorithm, as the surrogate perfectly models the task at hand. In Fig. 5, JES demonstrates empirically the value of the additional source of information, substantially outperforming PES and MES on all tasks.

Fig. 4 compares JES (top left) against PES, MES and EI in terms of point selection for one repetition on a two-dimensional sample task, where all runs are initialized with $D+1$ identical random samples. We observe that JES succeeds in finding all attractive regions of the search space, and queries the region around the optimum densely, which is sensible in a noisy setting. We further notice that EI (bottom right) fails to query the two circled local optima. PES (bottom left) also ignores two local optima to various degrees, and tends to circle the (perceived) optimum densely, which is expensive in terms of number of evaluations. We believe this showcases a shortcoming of only

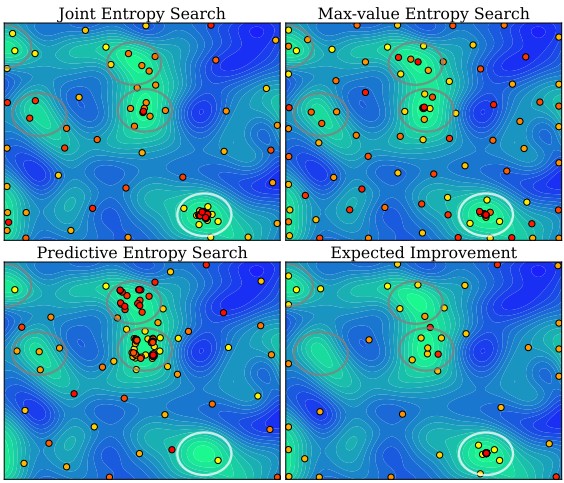

**Figure 4:** Comparison of queries for JES (top left), MES (top right), PES (bottom left) and EI (bottom right) on a sample of a 2D GP after a 100 function evaluations. The global optimum is circled in white, and four local optima in grey. Earlier queries are colored yellow, and later queries red.

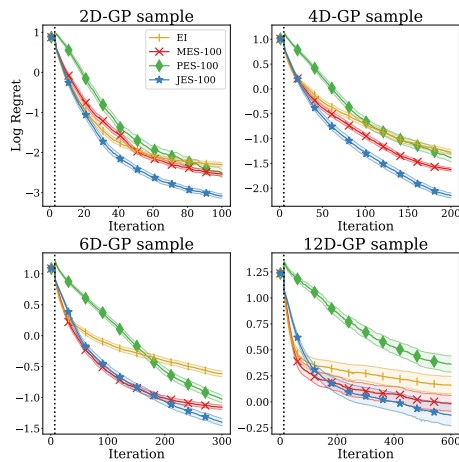

**Figure 5:** Comparison of JES, MES, PES and EI on GP prior samples. We run 1000 repetitions each for 2, 4 and 6D, and 250 on 12D. Mean and 2 standard errors of log regret are displayed for each acquisition function. The vertical dashed line shows the end of the initial design phase.

considering the density over the optimum: PES circles the optimum, but does not query its value. Lastly, MES (top right) successfully queries all attractive regions of the space, but also samples regions that are evidently poor the most densely out of the four approaches, despite information given by earlier (brighter) samples. Since JES considers the information conveyed by both MES and PES, it successfully excludes the apparent suboptimal regions of the space, finds all relevant optima, and queries these optima in a desirable manner.

We additionally evaluate the performance of all approaches on GP sample tasks that have a substantial amount of noise - its standard deviation roughly accounting for 10% of the total output range. We run these tasks with the GP hyperparameters fixed a priori for a larger number of iterations, 125D, to display the stagnation of some approaches. While MES and PES slow down approximately at the halfway point for both tasks, JES steadily improves for the entire length of the run. This robustness to large noise magnitudes highlights the importance of intrinsically handling noisy objectives in JES.

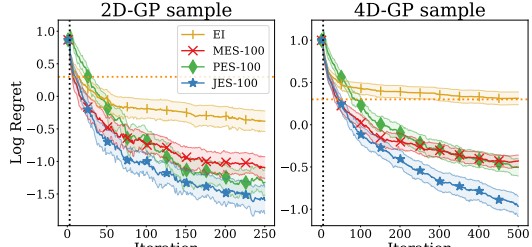

**Figure 6:** Evaluation of JES, MES, PES and EI on noisy ($\sigma_\epsilon^2 = 4$, orange) GP sample tasks across 100 repetitions. Mean and 2 standard errors of log regret are displayed for each acquisition function.

| Task | JES-100 | MES-100 | PES-100 | EI |
|------|---------|---------|---------|-----|
| 2D | $1.40 \pm 0.32$ | $1.03 \pm 0.19$ | $17.39 \pm 4.95$ | $0.23 \pm 0.13$ |
| 4D | $1.50 \pm 0.37$ | $1.21 \pm 0.3$ | $34.53 \pm 8.3$ | $0.3 \pm 0.17$ |
| 6D | $1.56 \pm 0.39$ | $1.26 \pm 0.37$ | $62.92 \pm 13.54$ | $0.35 \pm 0.2$ |

**Table 1:** Runtime of JES, MES, PES and EI on GP sample tasks of varying dimensionalities. JES is only marginally slower than MES, and orders of magnitude faster than PES.

In Table 1, we display the runtime of each acquisition function on these tasks when marginalizing over 10 sets hyperparameters, and sampling 10 optima per set. We time each iteration from after hyperparameters have been sampled, up until (but excluding) the query of the black-box function. Thus, acquisition function pre-computation and optimization are included. The runtime of JES is only marginally slower than that of MES with Gumbel sampling, while being at least an order of magnitude faster than PES for all displayed dimensionalities.

## 4.2 Synthetic test functions

Next, in Fig. 7, we study the performance of JES on three optimization test functions: Branin (2D), Hartmann (3D) and Hartmann (6D). For these tasks, we follow convention [16, 31] and marginalize over GP hyperparameters. On Branin, JES starts out slightly slower than MES but reaches the same performance in 100 iterations; and on the two Hartmann functions, JES performs amongst the best in the beginning and clearly best in the end. We note that PES experienced numerical issues on Branin, and as such, we acknowledge that its performance should be better than what is reported.

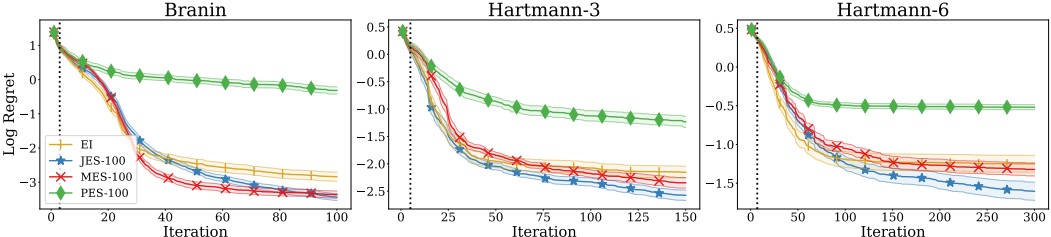

**Figure 7:** Comparison of JES, MES, PES and EI on Branin and Hartmann-6, $\sigma_n^2 = 0.10$. Mean and 2 standard errors of log regret are displayed for each acquisition function across 100 repetitions. The vertical dashed line represents the end of the initial design phase.

## 4.3 MLP tasks

Lastly, we evaluate the performance of JES on the tuning an MLP model's 4 hyperparameters for 20D iterations on six datasets. These tasks are part of the OpenML[1] library of tasks, and the HPO benchmark is provided through the HPOBench [10] suite. We measure the best observed classification accuracy. Notably, these tasks have a large amount of noise, which causes the performance to fluctuate substantially between repetitions. We observe that JES performs substantially better on two tasks, and is approximately equal in performance to EI on three, with EI being superior in one task. JES displays superior or equal performance to MES on all tasks, with PES lagging behind.

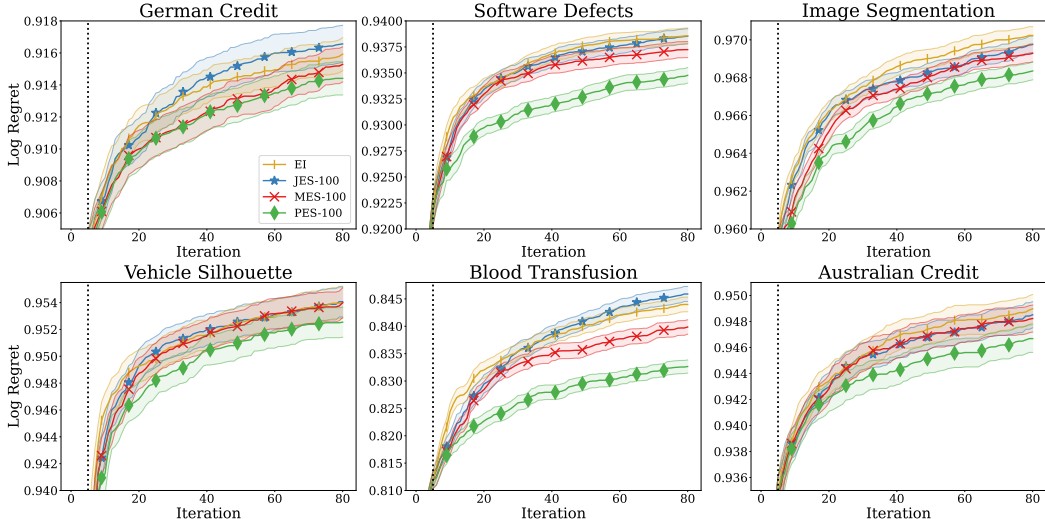

**Figure 8:** Comparison of JES, MES, PES and EI on six different MLP tuning tasks from the HPOBench suite. Mean and 1 standard error of best observed accuracy are displayed for each acquisition function across 100 repetitions. The vertical dashed line represents the end of the initial design phase.

---

[1] https://www.openml.org/

# 5  Conclusions

We have presented Joint Entropy Search, an information-theoretic acquisition function that considers an entirely new quantity, namely the joint density over the optimum and optimal value. By utilizing the entropy reduction from fantasized optimal observations, JES obtains a simple form for the entropy reduction regarding the joint distribution. As such, the additional information considered comes with minimal computational overhead, avoids restrictive assumptions on the objective, and yields state-of-the-art performance along with superior decision-making. We believe JES to be a new go-to acquisition function for BO, and to establish a new standard for subsequent information-theoretic techniques.

# 6  Limitations and Future Work

The main contribution of this paper is to provide a novel information-theoretic acquisition function which, given a sufficiently accurate model, yields impressive results. However, the non-myopic, speculative nature of information-theoretic approaches lend them to be susceptible to model misspecification, such as a poor choice of GP kernel or GP hyperparameters. In our view, information-theoretic approaches are possibly more susceptible to this issue than their myopic counterparts (EI, UCB, TS). While we propose a remedy to stabilize and improve the acquisition function under model misspecification with the $\gamma$-exploit approach, this technique only serves to *discover* misspecification and adjust accordingly, but not to inherently fix the misspecification. We believe misspecification can only be remedied by altering the surrogate model. It is thus very promising to combine advanced modelling techniques with information-theoretic acquisition functions, as already done with the additive GP approach utilized in conjunction with MES by Wang and Jegelka [47]; further promising additions would be to tackle heterogeneous noise and input warping as done by HEBO [8].

We also note that, since JES computes the entropy reduction from conditioning on the optimal pair, it relies on some level of noise in the objective. A surrogate model with zero noise will result in an infinite information gain for every optimal pair, which (by utilizing some random tie-breaking strategy) would make JES equivalent to TS. However, if JES is to be used in a completely noiseless setting, we argue that a small noise term should be added as a remedy. As this is done by default in many prominent GP frameworks [13, 14, 45], we do not view this as a major limitation of our approach. Nevertheless, improving upon this strategy would be interesting in future work.

For future work, we also envision work on the adaptation of JES to various different domains, such as multi-fidelity [48] and multi-objective optimization [1], as well as the integration of user prior knowledge over the location of the optimum [20] to accelerate optimization.

## Acknowledgments and Disclosure of Funding

Luigi Nardi was supported in part by affiliate members and other supporters of the Stanford DAWN project — Ant Financial, Facebook, Google, Intel, Microsoft, NEC, SAP, Teradata, and VMware. Carl Hvarfner and Luigi Nardi were partially supported by the Wallenberg AI, Autonomous Systems and Software Program (WASP) funded by the Knut and Alice Wallenberg Foundation. Luigi Nardi was partially supported by the Wallenberg Launch Pad (WALP) grant Dnr 2021.0348. Frank Hutter acknowledges support through TAILOR, a project funded by the EU Horizon 2020 research and innovation programme under GA No 952215, by the Deutsche Forschungsgemeinschaft (DFG, German Research Foundation) under grant number 417962828, by the state of Baden-Württemberg through bwHPC and the German Research Foundation (DFG) through grant no INST 39/963-1 FUGG, and by the European Research Council (ERC) Consolidator Grant "Deep Learning 2.0" (grant no. 101045765). The computations were also enabled by resources provided by the Swedish National Infrastructure for Computing (SNIC) at LUNARC partially funded by the Swedish Research Council through grant agreement no. 2018-05973. Funded by the European Union. Views and opinions expressed are however those of the author(s) only and do not necessarily reflect those of the European Union or the ERC. Neither the European Union nor the ERC can be held responsible for them.

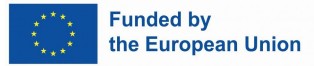

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
