# A   Broader impact

Our work proposes a novel acquisition function for Bayesian optimization. The approach is foundational and does not have direct societal or ethical consequences. However, JES will be used in the development of applications for a wide range of areas and thus indirectly contribute to their impacts on society. As an algorithm that can be used for HPO, JES intends to cut resource expenditure associated with model training, while increasing their performance. This can help reduce the environmental footprint of machine learning research.

# B   Experimental setup

**Frameworks.**   For all tasks and acquisition functions, we use the original PES implementation in MATLAB by Hernández-Lobato et al. [16], which uses the GPStuff [45] library. The implementation optimizes the acquisition function, and the posterior mean, by sampling a dense grid of points, and uses a gradient-based optimizer to further optimize the single best point. For better accuracy, we substantially increased the number of grid points.

**GP sample tasks.**   To generate the GP sample tasks, we use a random Fourier feature [29] with weights drawn from the spectral density of a squared exponential kernel. For dimension-wise length scale $\theta_d$, output scale $\sigma^2$, and noise variance $\sigma_\epsilon^2$, the hyperparameters per task are shown in Table 2. The range in the last column is a rough approximation of the magnitude of the output spanned by

| $D$ | $\theta_d$ | $\sigma^2$ | $\sigma_\epsilon^2$ | Approximate range |
|---|---|---|---|---|
| 2 | 0.1 | 10 | 0.01 | $[-9, 9]$ |
| 4 | 0.2 | 10 | 0.01 | $[-11, 11]$ |
| 6 | 0.3 | 10 | 0.01 | $[-13, 13]$ |
| 12 | 0.6 | 10 | 0.01 | $[-18, 18]$ |

**Table 2:** Hyperparameters for the generated GP sample tasks.

each GP sample. The length scales of the samples are gradually increased with each dimensionality to maintain a reasonable level of difficulty for all tasks. Since the optimal values for these tasks are unavailable, they are approximated through a dense random search, followed by local search on the most promising subset of points.

**Runtime tests.**   For the runtime tests, we run each acquisition function on 100 seeds on the 2, 4 and 6-dimensional GP sample tasks for 100 iterations each. To consider the runtime induced by each acquisition function in a realistic setting, we marginalize over 10 hyperparameter sets (including noise variance), and sample 10 optima each for JES, MES and PES. For these experiments, the runtime of an acquisition function is considered to be the time the time from after GP hyperparameters are sampled and the GP covariance has been inverted, until the query has been selected. Thus, only acquisition function setup and acquisition function optimization are considered as part of the runtime. All acquisition functions are optimized using an identical budget of 10000 raw samples, and a subsequent gradient-based optimization around the single best point.

**Synthetic test functions.**   For the synthetic test functions, 100 sampled optimal pairs are used for each acquisition function. GP hyperparameters are marginalized over for these tasks, so an equal number of optimal pairs are sampled for each hyperparameter set. The hyperparameters are re-sampled on a fixed schedule throughout the run. Naturally, the sampled maxima were updated at each iteration. Moreover, each test function was also given a fixed amount of noise. Regret was computed not from the noisy observed value, but from the true, noiseless function value.

**MLP classification tasks.**   All the classification tasks have a substantial amount of noise. As noiseless objective values are unavailable, we report the observed classification accuracy. 5 hyperparameters are available in HPOBench for these tasks. However, one of them (number of layers, *depth*) is held fixed due to its small integer-valued domain and the lack of integer hyperparameter support in the MATLAB framework. As seen in Tab. 4, the two other integer-valued hyperparameters *batch*

| Task | No. hyperparameter sets | No. maxima per set | Update frequency | $\sigma_\epsilon^2$ |
|---|---|---|---|---|
| Branin | 20 | 5 | 5 | 0.01 |
| Hartmann (3D) | 20 | 5 | 10 | 0.01 |
| Hartmann (6D) | 20 | 5 | 10 | 0.01 |
| MLP classification | 20 | 5 | 5 | 0.0 |

**Table 3:** GP Hyperparameter sets and updates for the synthetic test functions and MLP tasks.

| Name | Type | Range |
|---|---|---|
| Alpha (L2) | Continuous | $[10^{-8}, 10^{-3}]$ |
| Batch size | Integer | $[2^2, 2^8]$ |
| Depth | Fixed to 2 | $\{1, 2, 3\}$ |
| Initial learning rate | Continuous | $[10^{-5}, 1]$ |
| Width | Integer | $[2^4, 2^{10}]$ |

**Table 4:** Search space for the MLP tasks.

*size* and *width* have orders of magnitude larger domains, and can therefore reasonably be treated as continuous hyperparameters. All tasks are optimized in the [0, 1] range, and are scaled, transformed, and rounded to the nearest integer in the objective function where necessary. All non-fixed parameters are evaluated in log scale. The three tasks evaluated are *Australian*, *Blood-transfusion-service-center*, and *Vehicle*, with HPOBench task numbers #146818, #10101, #53, respectively.

**Compute resources.** All experiments are carried out on *Intel Xeon Gold 6130* CPUs. Each repetition is run on a single core. In total, approximately $50,000$ core hours are used for the experiments in the main paper, and an additional $20,000$ for the appendix.

## C   Ablation studies and model misspecification

We provide ablation studies for the hyperparameter controlling the ratio of exploitative selections $\gamma$. Moreover, we show how $\gamma$-exploit improves inference regret by recognizing regions believed to be optimal as poor. Lastly, we display the robustness of JES to the noise variance $\sigma_\epsilon^2$.

### C.1   Ablation studies

We provide an ablation study on $\gamma$ in terms of both the simple and inference regret of JES. While simple regret may be a more practically relevant metric, inference regret helps understand the ability of the acquisition function to successfully locate the optimum. Fig. 9 shows that $\gamma > 0$ improves simple regret, which is to be expected from its occasional greedy selection. However, as is shown in Fig. 10, a moderate fraction $\gamma \in \{0.05, 0.1\}$ also yields comparable or even improved inference regret on all tasks. Notably, $\gamma = 0.2$ yields slightly worse performance on Hartmann (6D), but yields marginally improved performance on Branin and Hartmann (3D). As such, the $\gamma$-exploit approach not only improves performance in terms of simple regret, but yields improved inference as well.

### C.2   Model misspecification

As mentioned in Sec. 3.5, the performance of information-theoretic methods can suffer substantially from model misspecification. In Fig. 11, we show for JES, PES and MES how the $\gamma$-exploit approach helps stabilize inference and improve inference regret, and yields substantially better simple regret for all methods. For Michalewicz (10D), we observe that the inference regret of MES gets substantially worse after iteration 150. With the $\gamma$-exploit approach, this issue is severely reduced. In Fig. 12, the corresponding simple regrets for MES deviate at iteration 150. The same behavior can be observed for JES on Levy (8D) around iteration 200. Across all test functions in Fig. 11 and Fig. 12, an $\gamma$-exploit strategy yields comparable or improved inference regret, and strictly improved simple regret for all acquisition functions. Moreover, we observe that MES is generally the top-performing acquisition function, implying that it is the most robust to model misspecification. Using the $\gamma$-exploit approach,

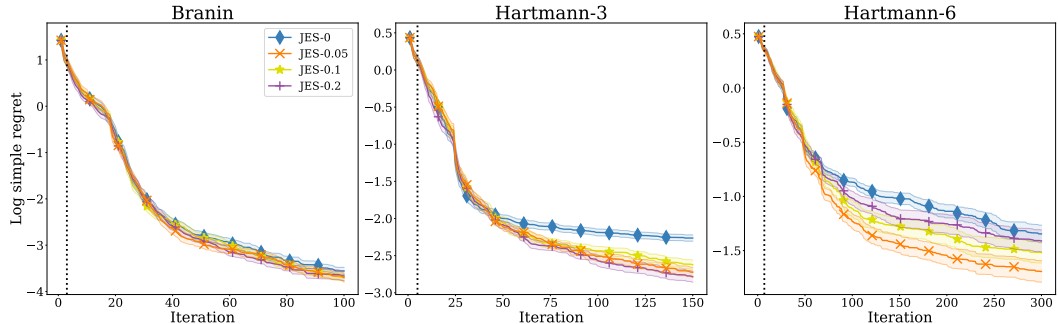

**Figure 9:** Comparison of JES with varying fraction $\gamma$ of exploitative selections. Mean and 1 standard error of log simple regret is displayed for all tasks.

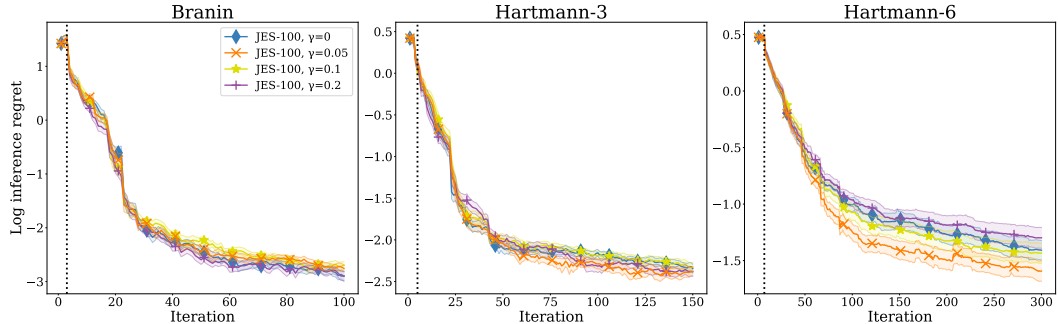

**Figure 10:** Comparison of JES with varying fraction $\gamma$ of exploitative selections. Mean and 1 standard error of log inference regret is displayed for all tasks.

the acquisition function verifies whether the belief over the location of the optimum is correct under the current model hyperparameters. If it is not, it re-calibrates its belief.

### C.3    Ablation study on $\sigma_\epsilon^2$

For the noise variance ablation study, we once again consider the GP sample tasks. We fix the GP noise hyperparameter $\sigma_\epsilon^2$ to the correct value prior to the start of the experiment. In Fig. 13, we show that the performance of JES is robust with respect to the noise level, while the performance of MES and PES decrease more drastically as the level of noise increases.

## D    Dependence on the number of MC samples

We show in Fig. 14 the dependence of JES on the number of MC samples for the GP sample tasks. JES displays a slight dependence on the number of GP samples, as initial performance improves marginally for larger number of samples. This is more prominent for higher-dimensional tasks, where a lower number of samples causes a substantially slower start. This can be explained by the fact that a larger number of sampled optimal pairs are required to accurately model a higher-dimensional density. Notably, the final regret on the 12-D benchmark is better for lower numbers of samples, such as JES-4. One potential explanation for this is its inability to model the joint distribution accurately. If all realized optimal pairs are close to the perceived optimum, JES is almost certainly going to sample there. Moreover, since the information gain for each sample is relatively low for samples in a well-explored region, the information gain of a few samples in an unexplored region may outweigh the information gain of a much larger number of samples in a well-explored region. For JES-4 and JES-20, it is more likely that all of the sampled optimal pairs are close to the optimum in this manner, while there is still small positive density on other parts of the search space. As such, it is possible that JES-4 and JES-20 over-exploit slightly in the 12-dimensional benchmark.

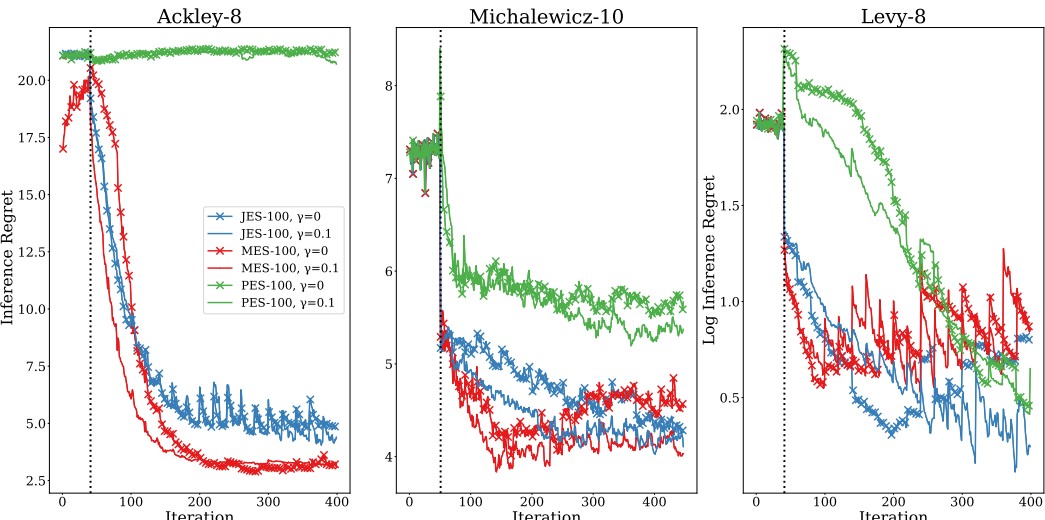

**Figure 11:** Mean of inference regret on high-dimensional synthetic functions for vanilla $\gamma$-exploit versions of JES, MES and PES. Upward spikes signify points where the GP hyperparameters are re-sampled, and the inference regret getting worse as a result. Error bars are omitted to increase legibility.

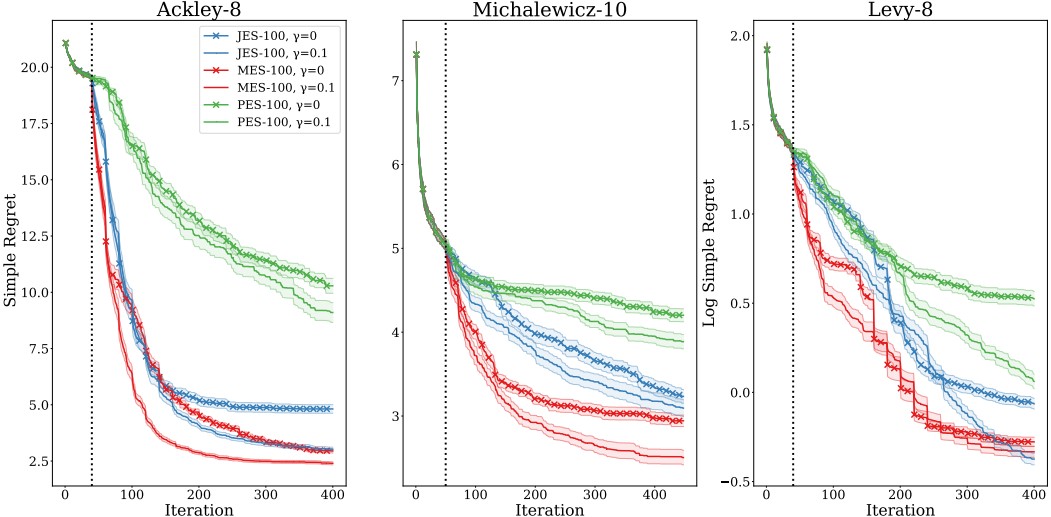

**Figure 12:** Mean and 1 standard error of simple regret on high-dimensional synthetic functions for vanilla $\gamma$-exploit versions of JES, MES and PES.

# E Approximation Quality

As stated in Sec. 3.4, we approximate the entropy of the posterior over observations conditioned on the data and the optimal pair $\mathrm{H}\left[p(y|\mathcal{D} \cup (\boldsymbol{x}^*, f^*), \boldsymbol{x}, f^*)\right]$ – the entropy of the sum of a Gaussian and a truncated Gaussian variable – by moment matching of the truncated distribution. We now show the quality of this approximation, and what impact it can potentially have on point selection. To do so, we utilize the results from Nguyen et al. [28] regarding the density of $p(y|\mathcal{D}, \boldsymbol{x}, f^*)$. We note that the approximation regards the *truncation* from knowing the optimal value $f^*$, which constitutes an additional reduction in entropy after having conditioned the GP on the new observation as in Sec. 3.3. As such, the approximation considers only a fraction of the total entropy reduction, as visualized in Fig. 3.

To establish the quality of the approximation, we compare our approach to approximating the entropy by MC. Naturally, the MC approach is more computationally expensive than moment matching, but

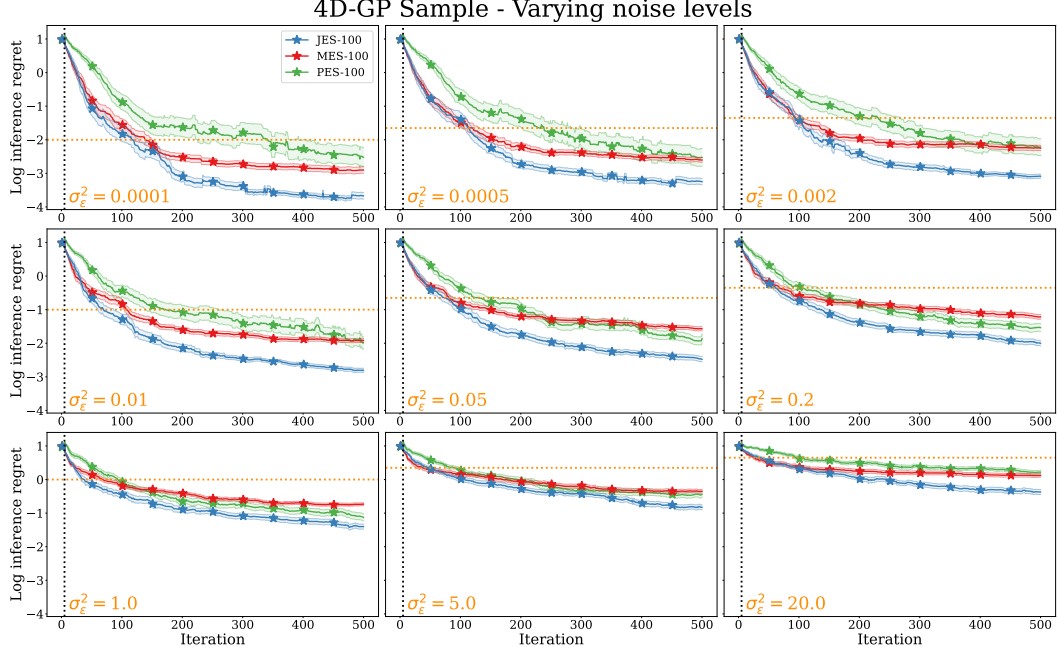

**Figure 13:** Evaluation of `JES`, `MES` and `PES` on noisy 4D GP sample tasks across 50 repetitions for 9 different noise levels. The noise variance $\sigma_\epsilon^2$ ranges from $10^{-4}$ (top left) to 20 (bottom right). Log noise standard deviation $\log(\sigma_\epsilon)$ is marked in dashed orange.

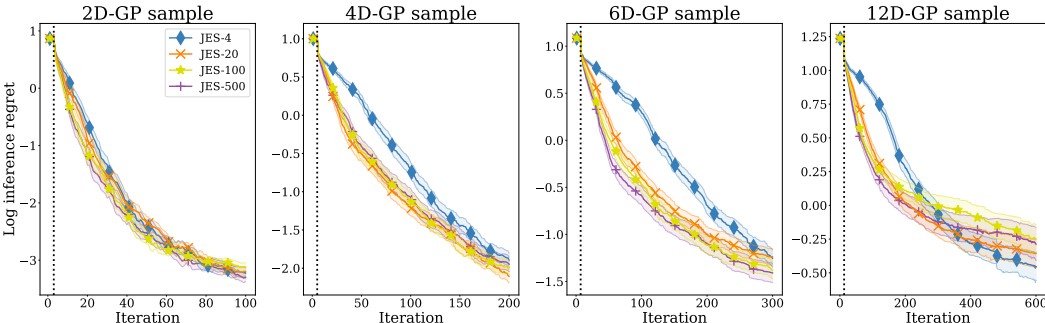

**Figure 14:** Comparison of JES with varying numbers of MC samples on GP tasks of varying dimensionalities - 10 repetitions on 10 unique tasks, for a total of 100 repetitions. Mean and 1 standard error of log inference regret is displayed for all tasks.

yields an asymptotically correct result. In Fig. 15, we show for varying noise levels, expressed as the noise variance ratio of the total variance, and truncation quantiles $\Phi^{-1}(\alpha)$, the difference in entropy between an MC and a moment matching approach. For example, a truncation quantile of $10^{-2}$ means that the upper $99\%$ of the density of the posterior distribution is removed as a result of truncation. We see that the approximate entropy reduction from moment matching is consistently lower than for the asymptotically exact MC approach. Moreover, the approximation error seemingly increases logarithmically with the truncation quantile. As such, we can expect modest underestimates of the entropy reduction when we truncate the posterior to an extreme degree, and the level of noise is low. In the right image of Fig 2, the posterior is severely truncated left of the conditioned location. However, the noise variance constitutes a large fraction of the total variance at this location, which means that the approximation is still accurate with respect to the true entropy reduction. The scenario represented in the bottom left corner of the grid, where we severely truncate the posterior *and* the noise variance is low, does not reasonably occur in practice. The aforementioned scenario would entail sampling an optimal pair which is several orders of magnitudes *worse* than the mean of the GP at uncertain (in the sense that $\sigma >> \sigma_\epsilon$) locations. Lastly, the blue region in the upper left corner of

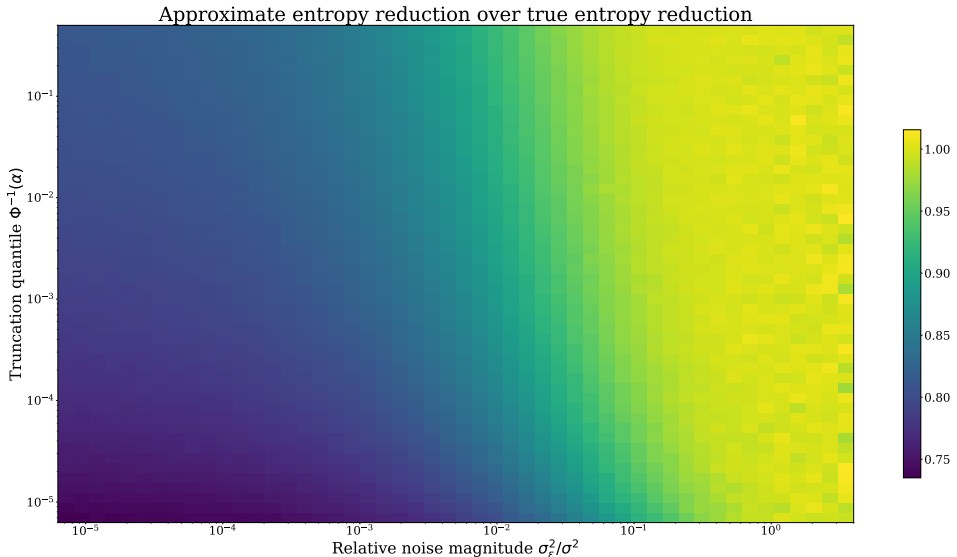

**Figure 15:** Visualization of approximation error from the moment matching approach compared to an asymptotically exact MC approach. The colormap represents the fraction of the entropy reduction resulting from truncation as approximated from moment matching divided by the entropy reduction as computed through MC. The inconsistencies in coloring in the rightmost part of the image are caused by inconsistent MC approximation.

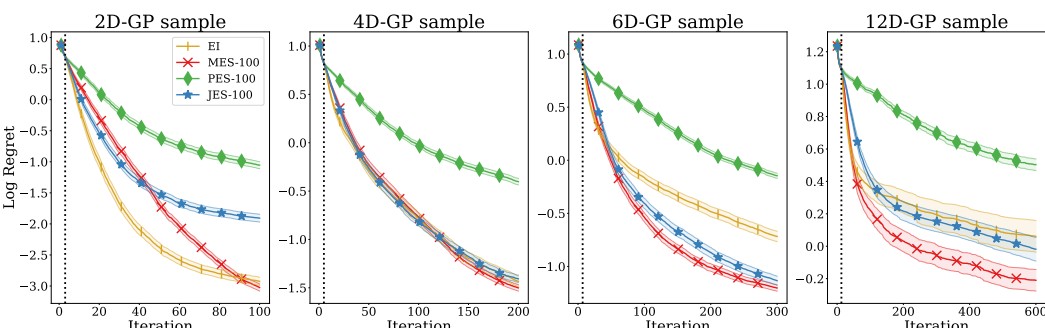

**Figure 16:** Comparison of `JES`, `MES`, `PES` and `EI` on GP prior samples using simple regret. We run 1000 repetitions each for 2, 4 and 6D, and 250 on 12D. Mean and 2 standard errors of log regret are displayed for each acquisition function. The vertical dashed line shows the end of the initial design phase.

the grid represents a region where we have less truncation, and the noise is a relatively insignificant part of the total variance. Fig. 15 shows that the entropy reduction from truncation in this region is underestimated by approximately 15%. As such, the approximation error leads to a slightly less explorative strategy than what a strategy with exact computation of the truncation term would provide.

# F   Regret Measures

We display the regret measures for the GP sample and synthetic test functions.

## F.1   GP sample tasks

In Fig. 16, we show the simple regret for the GP sample tasks. We note that the simple and inference regrets for `MES` are approximately equal, while there is a substantial difference for `JES`. For `PES`, the difference between simple and inference regret is the most pronounced at approximately two order of magnitude at the most.

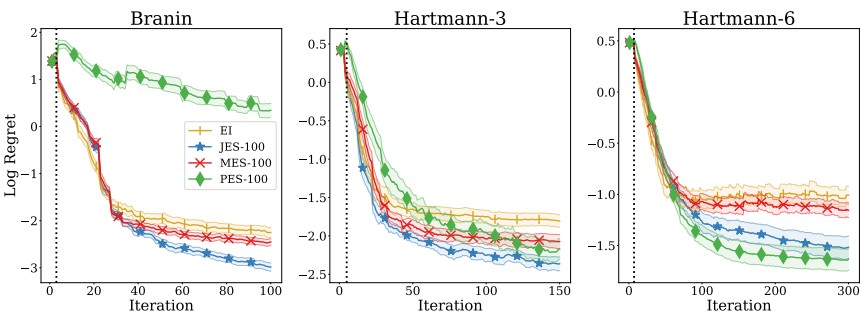

**Figure 17:** Comparison of JES, MES, PES and EI on Branin and Hartmann-6, $\sigma_n^2 = 0.10$. Mean and 2 standard errors of log regret are displayed for each acquisition function across 100 repetitions. The vertical dashed line represents the end of the initial design phase.

## F.2   Synthetic test functions

Next, we show the inference regret for the synthetic test functions in Fig. 17. We note that he simple and inference regrets for JES are approximately equal. For PES, the difference between simple and inference regret is once again very pronounced. Notably, the simple regret is significantly better than the inference regret for MES on Branin, implying that it does not yet have full knowledge of where the optimum is. We once again note the numerical issues of PES on Branin. Moreover, the inference regret of MES gets marginally worse for Hartmann (6D) from approximately iteration 100 until the end of the run, implying that its knowledge about the location of the optimum gets worse.