# OpenReview forum: "Joint Entropy Search For Maximally-Informed Bayesian Optimization"
_NeurIPS.cc/2022/Conference — NeurIPS 2022 Accept_

### Official Review · Reviewer_G9Zj · 2022-07-02

**Rating:** 8
**Confidence:** 3
**Soundness:** 4 excellent
**Presentation:** 4 excellent
**Contribution:** 3 good

**Summary:**

The authors propose Joint Entropy Search (JES), an acquisition strategy for Bayesian optimization (BO) that combines ideas from entropy search and max-value entropy search while retaining low computational complexity. As ingredients for this approach, the authors require a sampling scheme for potential candidates of optimum/optimal value pairs (i.e., f* and x* at which f*=f(x*)) and a method to compute the entropy of a truncated distribution. They also propose a scheme that prevents stalling in case the surrogate model is misspecified. The method is shown to outperform previously proposed information-theoretic acquisition strategies for BO.

**Questions:**

1) How are (4) and (5) equivalent? The mutual information terms are clearly different.
2) More generally, in (4)-(6) it is not fully clear what objects are random and what are deterministic, i.e., instances. E.g., in (4) the term $I((x,y);x^*|\mathcal{D})$ suggests, in a strong information-theoretic understanding, that all objects are random. I assume, however, that $\mathcal{D}$ and $x$ are instances or fixed constants, while $y$ and $x^*$ are random. ($y$ being the random output associated with the fixed input $x$.) This ambiguity, which is due to the common abuse of notation, makes these equations a bit difficult to parse.
3) In line 101 the mutual information objective of MES differs from the one in (6). Which is correct?
4) In (10), the argument of the first expectation appears to be independent of $b$, while the argument of the second clearly is; how can this be reconciled?
5) The entropy of a truncated normal distribution is not much more complicated than the entropy of a normal distribution. For the sake of clarity, I would suggest to apply moment matching not to the truncated Gaussian distribution, but to the sum $y=f+\epsilon$. The final result is the same, but this way the approximation is applied to the object that has no tractable entropy, rather than to the object that has.
6) In Algorithm 1, what is the meaning of $y_n=f(x_n)+\epsilon$? Does that mean that a random $y_n$ is sampled?
7) In the noisy setting in Fig. 6, why is PES slowing down together with MES? The noise-free assumption is inherent in MES, but it is less clear what the impact of noise is on PES. A few lines of text would clarify that, which would also explain the suboptimal performance of PES in Fig. 8.
8) In the experiments, ES was not used as a competing method. I assume that this is related to the fact that usually higher-dimensional entropies need to be evaluated than for PES. Nevertheless, it would be interesting to hear the authors opinion on what they believe would be the performance of ES.

**Limitations:**

The authors acknowledge the limitations of their approach (performance under model misspecification and the requirement of a noisy surrogate model).

**Strengths And Weaknesses:**

The paper is well-written. The pictorial illustration of the method is exemplary and illuminating. The experimental evidence is convincing. The only possible critique I see is that the approach itself appears to be obvious and incremental, being a combination of existing approaches. On the other hand, the authors successfully show that their combination of techniques not only improves performance, but also reduces the computational complexity compared to existing approaches (ES and PES). This critique is, therefore, not a major weakness and I am very much in favor of the manuscript.

My only substantial suggestion is to accompany the theoretical section with a few explanations on why JES has the potential to outperform ES/PES/MES. For example, it is claimed that MES relies on the noiseless setting; what ingredients make JES robust in the noisy setting, then? Also, PES is said to be computationally expensive; why can't the computational approach of JES be utilized to reduce the computational complexity of PES (e.g., extending lines 177-181)?

I see a few minor weaknesses that can easily be resolved by answering the questions below. The following minor comments may help to further improve the quality of the manuscript:
- l21: "modeling the unknown objective". Should "objective" be replaced by "black-box function"? I would expect that not the objective is modeled by the surrogate, but the black box.
- l28: space missing around "PES"; also please check consistent typesetting of ES and PES.
- l79: "employ": in my understanding, acquisition functions do not employ, but rather define this trade-off.
- eq. (3): $y^*_{n+1}$ should be replaced by $y$, or even better by $f(x)$, since we assume a noiseless function.
- The usage of (1) and (2) in lines 130-136 is slightly confusing, as I was briefly looking for the presumably referenced equations.
- Algorithm 1: lin 5 uses $\mu$ and $s^2$, while I assume the correct notation would be $m_n$ and $s_n$; in line 10, $y^*_\ell$ should be replaced by $f^*_\ell$; in line 13, the subscript $JES$ should be in typewriter font.
- l242 claims that "half an order of magnitude" is gained except for 12D, while Fig. 5 suggests that this also does not happen for 6D.

---

> ### Author Response · Authors · 2022-07-31
> **Response to Reviewer G9Z (2/2)**
>
> ### 4. Additional questions
>
> > ##### How are (4) and (5) equivalent? The mutual information terms are clearly different.
>
> This follows from the symmetric property of the mutual information. The reviewer is referred to the PES paper [15], where this symmetry transformation was first introduced, for further details.
>
> > ##### More generally, in (4)-(6) it is not fully clear what objects are random and what are deterministic, i.e., instances…
>
> The reviewer is correct in that $\mathcal{D}$ and $x$ are deterministic, and that there may be a more precise notation for these expressions. We have chosen to adapt the notation of previous work [15, 16, 17, 26, 29, 39, 43] for consistency.
>
> > ##### In line 101 the mutual information objective of MES differs from the one in (6). Which is correct?
>
> Thank you for bringing this to our attention! Line 101 is missing the conditioning on the data, which is $I((x; y); y^*|\mathcal{D})$. We added this missing piece in the new pdf.
>
> > ##### In (10), the argument of the first expectation appears to be independent of $b$, while the argument of the second clearly is; how can this be reconciled?
>
> The expectation over $b$ in the first expectation is a typo; we removed it in the new pdf. Thanks for pointing this out.
>
> > ##### The entropy of a truncated normal distribution is not much more complicated than the entropy of a normal distribution. For the sake of clarity, I would suggest to apply moment matching not to the truncated Gaussian distribution, but to the sum…
>
> Thanks for the suggestion. We will consider this formulation.
>
> > ##### In Algorithm 1, what is the meaning of $y_n = f(x_n) + \epsilon$? Does that mean that a random $y_n$ is sampled?
>
> This means that we acquire the noisy observation $y_n$ from querying the objective function at $x_n$. If you have a suggestion that would make this clearer, we would be happy to hear it.
>
> > ##### In the noisy setting in Fig. 6, why is PES slowing down together with MES? The noise-free assumption is inherent in MES, but it is less clear what the impact of noise is on PES…
>
> Our explanation for why PES performs relatively poorly in Fig. 8 is due to its inability to query greedily, rather than the impact of the noise. We refer the reviewer to Fig. 4 (bottom left) for a detailed visualization of how PES does not necessarily query the regions it believes are good, but rather circle them.
>
> For details as to why PES slows down in Fig. 6, we hypothesize that this could be due to constraint C2 in the original PES paper. In short, C2 is required to approximately condition on $x^*$, and the quality of the approximation varies with the noise level of the objective.
>
> > ##### In the experiments, ES was not used as a competing method (...) it would be interesting to hear the authors opinion on what they believe would be the performance of ES.
>
> We did not compare against ES because PES is generally at least as good as ES, with few exceptions, as studied in [15, 43]. This is consistent with recent work [26, 29] where ES was omitted. In addition, an implementation of ES is no longer available (the original repository has broken dependencies).
>
> We again thank the reviewer for the helpful review and the appreciation of our work. We hope to have sufficiently answered your questions, and if not, are happy to address any further questions you have about our work.

---

> > ### Comment · Reviewer_G9Zj · 2022-08-04
> > **Thanks!**
> >
> > Thank you for your responses and explanations, and for updating Sec. 3.1.
> >
> > I now understand that the apparent difference between (4) and (5) is due to the used notation. For example, it is not clear to me how $I((x,y);x^*)=I(y;(x,x^*))$; they would actually be different if $x$, $y$, and $x^*$ were random variables. Since $x$ is a constant, there may be some arbitrariness of putting it in a tuple with $y$ or $x^*$, respectively. This is what I was referring to. (And this happens again in the text surrounding (6), where $x$ is in a tuple with $y$, while in (6) it is in a tuple with $y^*$.)

---

> > > ### Author Response · Authors · 2022-08-05
> > > **Further response to Reviewer G9Zj**
> > >
> > > We once again thank the reviewer for offering improvements to our paper. Moreover, we agree that the use of tuples make the aforementioned expression less clear. Disregarding the data $\mathcal{D}$ in Eq. (4-6), $I(x^*; y|x) = I(y; x^*|x)$ would arguably have been a better notation, since it conveys that the symmetry property is with regard to  $x^*$ and $y$, and that $x$ is fixed. However, it would be suboptimal to change the widely adopted notation.
> > >
> > > Eq. (6) ultimately condiders a different quantity than Eq. (4) and Eq. (5), The same reformulation is applied, but now with regard to $y$ and $y^*$ instead of $y$ and $x^*$. As such, the step that is left out in Eq. (6), by the suggested notation, is $I(y^*; y|x) = I(y; y^*|x)$.

---

> > > > ### Comment · Reviewer_G9Zj · 2022-08-05
> > > > **Thanks again!**
> > > >
> > > > This clarifies things perfectly, thanks a lot! I agree that changing notation may not be necessary, given a general understanding in the field.

---

> ### Author Response · Authors · 2022-07-31
> **Response to Reviewer G9Z (1/2)**
>
> We thank the reviewer for the valuable feedback, and for appreciating the approach and the general quality of our paper. We have addressed your questions and critiques in our responses below, where some of your comments have been split up for clarity. A new version of the paper has been uploaded, which incorporates the suggested changes.
>
>
>
> ### 1. Benefit of Joint Entropy
>
> > ##### My only substantial suggestion is to accompany the theoretical section with a few explanations on why JES has the potential to outperform ES/PES/MES.
>
>
> The fundamental reason for why JES can outperform PES and MES is due to the additional information that is considered. In the case of MES, the task of reducing the entropy regarding the optimal value arguably serves as a proxy for the entropy over the optimum. This proxy is evidently a very good one, but a proxy nonetheless.
>
> _Now, why would the information regarding $f^*$ help when considering the information gain regarding $x^*$, like in ES/PES?_
>
> Finding $x^*$ _and_ $f^*$ may be more challenging than finding only $x^*$. However, there is a tradeoff where the information that we gain regarding $f^*$ (for instance that the maximum is likely below some upper bound C) may help us in finding $x^*$ quicker. Acquiring information about the likely range of $f^*$ helps in determining which regions are more probable of containing $x^*$.
>
> We have added such an explanation in Sec. 3.1 of the new version of the paper.
>
>
>
> ### 2. Robustness to noise
>
> > ##### For example, it is claimed that MES relies on the noiseless setting; what ingredients make JES robust in the noisy setting, then?
>
>
> There are two reasons for this:
>
> 1. JES considers the noiseless optimal value $f^* $as opposed to a maximal value $y_{max}$, because $y_{max}$ yields an overestimated information gain as explained by [39].  This is also convenient because the distribution over the noisy outputs $y$  do not follow a truncated normal distribution when we condition on $f^*$, as shown by [26, 39] and Moss et. al., (2021).
> 2. The information gain over the optimal value is substantially reduced when there is large noise, which is why it is beneficial to consider both $x^*$ and $f^*$ in this setting.
>
>
>
> ### 3. Utilizing JES to speed up PES
>
> > ##### Also, PES is said to be computationally expensive; why can't the computational approach of JES be utilized to reduce the computational complexity of PES (e.g., extending lines 177-181)?
>
>
> To compute the information gain, one needs to condition on the relevant quantity, e.g., $x^*$ in the case of PES. Conditioning only on the location of the optimum, without a corresponding observed optimal value, is not intrinsically supported by a GP, which is why PES needs to condition on other types of observations, e.g., a negative definite Hessian at $x^*$, to accomplish this. Since we condition on $(x^*, f^*)$, we have a fantasized data point, which is something that the GP can naturally condition on.
>
>
>
>
> ## References:
> GIBBON: General-purpose Information-Based Bayesian OptimisatioN. HB Moss, DS Leslie, J Gonzalez, P Rayson. _Journal of Machine Learning Research_ (JMLR). 2021.

---

### Official Review · Reviewer_ms2M · 2022-07-11

**Rating:** 6
**Confidence:** 4
**Soundness:** 3 good
**Presentation:** 4 excellent
**Contribution:** 2 fair

**Summary:**

The main idea of the proposed acquisition function in this work is to consider the information gain between evaluation point and both the optimum x* and optimal value f* of the black-box function; in some sense combining PES and MES that was proposed in the past. The new acquisition function samples functions from the posterior GP and optimize each one to obtain samples of x* and f*, which does not use more computational cost comparing to PES.

**Questions:**

How much of the empirical gains of JES is because of the greedy selection approach in section 3.5? What does the comparison to baselines look like if JES does not use \gamma? Does using \gamma values still help if there's no model misspecification?

Optimizing sampled functions in the inner loop Thompson sampling step might lead to suboptimal x* and f*, especially for high dimensional functions. How much does this inner loop optimization matter to JES? Do we have any comparisons on different inner loop optimizers to show the sensitivity?

**Limitations:**

The authors mentioned model misspecification and lack of analysis on high dimensionality, multi fidelity etc as limitations.

I think some more important limitations that don't seem to be mentioned is the stability of the method which depends on good inner loop optimization, good selection of \gamma, approximation errors and lack of theoretical guarantees. I think those need to be addressed in this work more systematically.

**Strengths And Weaknesses:**

Strengths:
- This is a clever idea to combine MES and PES, given that the Thompson sampling step will produce both x* and f*. PES only uses x* and MES only uses f*. Using both will in fact incorporate more information.
- The clarity of this work is very good. The motivation and most steps of the method are explained very well.
- Another interesting point is the inverse \gamma greedy approach. It's a simple and nice addon that can be used for all entropy search methods, whose goals are not evaluating the optima.

Weaknesses / suggestions
- The most critical component of the proposed acquisition function is the computation of \sigma_{f|f*} but the main paper doesn't seem to talk about how it is computed. I think this results in a loss of main contribution of this work and would suggest focusing on how this quantity is computed in the main paper.
- This work might have some issues if f has multiple optima since it ignores regions of good x* from the same sampled function, which could be the problem of PES too. Also note that optimizing sampled functions in the Thompson sampling step might lead to suboptimal x* and f* depending on how good the global optimizer is. This will also lead to potential stability issues of the proposed method.
- While the method is very interesting, it also feels like an incremental step from PES and MES. The empirical gains on realistic tasks in section 4.3 doesn't really show much improvement comparing to baselines. Neither is there theoretical gains comparing to past methods. This problem is not easy to resolve. The next comment may shed some lights on this.
- The inverse \gamma greedy approach feels more like a heuristic in the current presentation. I think it's possible to further improve the presentation by analyzing why this approach is needed in entropy search methods. I think Figure 9 is a nice way towards better understanding. Some theoretical insights will also help this approach to be more principled, e.g. how to select \gamma. I think the authors can benefit a lot from making this greedy selection a highlight of this work (once we have some principled analysis). Doing so will also avoid criticisms on being too incremental.

---

> ### Author Response · Authors · 2022-07-31
> **Response to Reviewer ms2M**
>
> We thank the reviewer for the thorough feedback. We address the reviewer's comments below and we have uploaded a new version of the paper on OpenReview.
>
>
> ### 1. Computation of $\sigma_{f|f*}$
>
> > ##### The most critical component of the proposed acquisition function is the computation of $\sigma_{f|f*}$ but the main paper doesn't seem to talk about how it is computed…
>
>
> The computation of $\sigma_{f|f*}$ is described in Sec. 3.6, line 206 in the old version as the variance of a truncated Gaussian, with the optimal value and the GP mean and variance as parameters. We have now moved this to Sec. 3.4, Eq. (11) to make this point clearer in the new pdf.
>
> If the reviewer still feels that this computational step warrants more details, we would be happy to provide them.
>
>
>
> ### 2. Issues of multiple optima
>
> > ##### This work might have some issues if f has multiple optima since it ignores regions of good $x^*$ from the same sampled function, which could be the problem of PES too. Also note that optimizing sampled functions in the Thompson sampling step might lead to suboptimal $x^*$ and $f^*$ depending on how good the global optimizer is…
>
>
> We are unsure why the reviewer believes that JES would have issues if f has multiple optima, as we are not sure what is meant by “ignoring regions of good $x^*$”. Could you please elaborate on this comment? In Fig. 4, we provide an example of a highly multimodal function, where JES successfully finds all competitive modes.
>
> We acknowledge that our method is contingent on the Thompson sampling being reasonably accurate. However, also true for PES, and TS itself. MES on the other hand, is contingent on sufficiently (but not overly) covering the search space for the Gumbel approximation. As the information in JES will likely be largest around the sampled optima, JES drastically simplifies the optimization of the acquisition function, effectively moving the difficulty of optimization to the sampling of optima.
>
>
>
> ### 3. Incrementality of JES
>
> > ##### While the method is very interesting, it also feels like an incremental step from PES and MES. The empirical gains on realistic tasks in section 4.3 doesn't really show much improvement comparing to baselines…
>
>
> We believe that our novel acquisition function, with an innovative, yet simple approach to computing information gain, yields sufficient novelty. Moreover, the novel formulation yields surprisingly simple computations, which all but removes the need for costly approximations as in the case of PES. The simplicity of the approach is a big advantage. For a decade, information-theoretical acquisition functions have been looked at with great interest by the BO community. However, to date they are rarely adopted in practice and both ES and PES are considered difficult both to implement and understand. We hope that our method will lead to a democratization of the techniques that model a distribution over $x^*$ by providing a simpler yet demonstrably effective method - one with the conceptual and computational complexity of MES. This can be associated with the natural evolution of a piece of research artifact into a technology, i.e., robust and simple to use.
>
> Furthermore, our introduction of the inverse-gamma greedy approach is also novel.
>
>
>
> ### 4. Heuristic nature of inverse-$\gamma$
>
> > ##### The inverse $\gamma$ greedy approach feels more like a heuristic in the current presentation. I think it's possible to further improve the presentation by analyzing why this approach is needed in entropy search methods (...) Some theoretical insights will also help this approach to be more principled…
>
>
> We appreciate this great piece of feedback, and agree that this contribution could be featured more prominently. The suggestions that you have outlined are very valuable, and will be taken into account to improve our work.
>
> We thank again the reviewer for the helpful review and we hope that our answers helped clarify the reviewer's concerns. If we have successfully done so, we would appreciate it if the reviewer would consider increasing the score.

---

> > ### Comment · Reviewer_ms2M · 2022-08-05
> > **Response**
> >
> > Appreciate the author's reply and clarification. I agree multi optima is probably not a problem for JES.
> >
> > However, I found this statement confusing: "information-theoretical acquisition functions have been looked at with great interest by the BO community. However, to date they are rarely adopted in practice...". Note that MES was implemented in BoTorch.
> > https://botorch.org/tutorials/max_value_entropy
> >
> > Also, the authors seemed to have missed my questions under "Questions:".

---

> > > ### Author Response · Authors · 2022-08-08
> > > **Additional response to Reviewer ms2M**
> > >
> > > We once again thank the reviewer for the initial feedback. Our apologies for missing the additional questions.
> > >
> > > > #### I found this statement confusing: "information-theoretical acquisition functions have been looked at with great interest by the BO community. However, to date they are rarely adopted in practice...". Note that MES....
> > >
> > > The top performing information-theoretic method, MES, is unfortunately not commonly used by practitioners. EI is chosen as a default method by framework developers as explained in the table below:
> > >
> > >
> > > | Popular frameworks in BO  | MES available | MES chosen as default acq. function |
> > > |---------------------------|---------------|--------------------------------------------|
> > > | Ax/BoTorch                | Yes           | No                                         |
> > > | SMAC3                     | No            | No                                         |
> > > | Dragonfly                 | No            | No                                         |
> > > | Trieste (prev. GPFlowOpt) | Yes           | No                                         |
> > > | SyneTune                  | No            | No                                         |
> > > | GPyOpt                    | No            | No                                         |
> > >
> > > Given the new evidence that JES brings, we expect that the perception of the BO practitioners and framework developers towards information-theoretic acquisition functions will change.
> > >
> > > > #### Optimizing sampled functions in the inner loop Thompson sampling step might lead to suboptimal x* and f*, especially for high dimensional functions. How much does this inner loop optimization matter to JES? Do we have any comparisons on different inner loop optimizers to show the sensitivity?
> > >
> > > We observed that JES is robust w.r.t. a simple inner loop optimization method, e.g., an increase by a factor of 2 in the amount of grid samples doesn't lead to a substantial improvement in performance. However, we did not further study if adopting a different method (such as a gradient-based optimizer or substituting the random Fourier feature approximation with other types of posterior approximations) would improve the results; we leave this for future work. We thank the reviewer for pointing this out and we are excited about the opportunity of a potential further improved performance by adopting a more sophisticated treatment of the posterior samples for JES.
> > >
> > > > #### Does using $\gamma$ values still help if there's no model misspecification?
> > >
> > > Figures 4 and 5 study the case where no model specification is present. These examples constitute an optimal setting for each algorithm, as the GP surrogate perfectly models the task at hand by construction. In that setting, there is no intrinsic benefit of using the inverse-$\gamma$ strategy, since we know that JES' joint density over the optimum is correct, i.e., since the black-box function is a sample from the current GP, its optimum is distributed according to the modeled density. So, in the case of no model misspecification, the best strategy is to consider the maximal information gain at every iteration, and to select the argmax of the posterior mean as the final query — JES for $N-1$ iterations, greedy for iteration $N$, since we know that we have modeled the black-box function correctly. However, we argue that a perfect surrogate is very unlikely in practice, and while Fig. 4 and 5 show the intrinsic efficiency of JES, they are an unrealistic setting in practice.
> > >
> > > > #### How much of the empirical gains of JES is because of the greedy selection approach in section 3.5? What does the comparison to baselines look like if JES does not use $\gamma$?
> > >
> > > The GP sample tasks in Section 4.1 are intended to display the efficiency of JES without any confounding factors — No model misspecification, $\gamma=0$, and evaluation on inference regret (which conveys the accuracy of the model’s belief over where the optimum is). Here, JES is significantly superior to MES, PES and EI on all tasks. Due to the lack of confounding factors, we believe this result is remarkable.
> > > In Section 4.2 onwards, we consider a more practical setting — Possible model misspecification, and evaluation on simple regret. In Appendix C.1, we show the sensitivity of JES to the choice of $\gamma$ on all the synthetic test functions in terms of inference regret and simple regret. In particular, Fig. 10 shows that JES with $\gamma=0$ is approximately on par with $\gamma>0$ variants in terms of inference regret and  $\gamma=0$ performs worse in terms of simple regret as shown in Fig. 9. However, $\gamma=0$ is substantially worse for JES, MES and PES for high-dimensional tasks as shown by the simple regret in Fig. 12 where the model mismatch is emphasized by the inference regret in Fig. 11.
> > >
> > >
> > > We thank the reviewer for the feedback and continuous engagement. We hope to have sufficiently answered your questions, and if not, are happy to address any further questions you have about our work.

---

### Official Review · Reviewer_7H23 · 2022-07-11

**Rating:** 3
**Confidence:** 3
**Soundness:** 2 fair
**Presentation:** 2 fair
**Contribution:** 2 fair

**Summary:**

The paper proposes a variant of entropy search for Bayesian optimization, which considers joint entropy of the optimal solution x* and the optimal value f*. The approximation for the entropy and the treatment of observation noise are mainly discussed. Further, a heuristics for model misspecification is also introduced. The performance is evaluated on GP prior functions, well-known benchmark functions, and an MLP optimization.

**Questions:**

In the experiment, what does 'end of the initial design' mean? Why is performance of each method is different at that point? (e.g., in Fig 5 and 7)

**Limitations:**

The limitations are show in Section 6.

**Strengths And Weaknesses:**

The paper proposes another variant of entropy search.

+ The intuition behind the basic idea is simple.
+ A variety of empirical evaluations are provided.

However, throughout the paper, the technical descriptions are fully misleading and inaccurate. The authors sometimes ignore techniques in the past work (nevertheless it is cited), which unfairly exaggerates own contributions. Many inaccurate descriptions hide approximations of the entropy computations in the proposed method, by which readers can misunderstand that the proposed method performs more accurate computations than it actually performs. Detailed comments are as follows.

- Although the authors implicitly assumes conventional MES is only for noiseless, it is not the case. [22] and [39] have already shown how to incorporate the additive noise. Note that although [39] is for multi-fidelity optimization, it contains the usual single-fidelity setting as a special case (For the detail of the noise treatment, I'll mention later again). Therefore, the description of the third main advantage listed in the end of introduction is misleading because readers would think that other information approaches cannot handle the observation noise. However, the observation noise can be handled by the above literature, and further, PES [15] can also incorporate the additive observation noise.

- From (8) to (9), p(y|D,x,x*,f*) is re-written as p(y|D U (x*,f*),x,f*). After that, in Section 3.4 (which describes the computations of the entropy of p(y|D U (x*,f*),x,f*))), it is further re-written as p(y|D,f*). This re-writing hides the fact that the original p(y|D,x,x*,f*) contains a condition that 'x* is the maximizer'. For example, in the case of PES [15], a condition that the hessian of x* should be negative definite is considered. This condition is implicitly ignored by the proposed method (without mentioning anything about this approximation), and the notation p(y|D,f*) makes difficult to be aware of that fact.

- The proposed method incorporates (x*,f*) into GP to calculate p(y|D,f*), but f(x) <= f* is only (approximately) imposed for the current queried x. Note that f(x) for other x in \cal{X} is not constrained in the proposed method (though f(x) <= f* is originally imposed on all x). Similar approximations have been employed by past works, but in the paper, this approximation is implicitly introduced, by which readers can mistakenly consider that the calculation in the proposed method is exact computations unlike previous ones.

- About MES, the authors described that MES considers y* = f(x*) + \epsilon (according to the descriptions after (6)). However, this is not the case. In the original MES paper [43], y* is defined as f(x*) (noise is not included in y*). Therefore, although the authors claim the replacement y* with f* is one of differences from MES after (9), it is also misleading.

- About MES, in line 103-104, the authors described

> p(y|D, x, y∗) is a truncated Gaussian distribution, for which the entropy can be computed in closed form. However, p(y|D, x, y∗) takes this form only in a strictly noiseless setting

This statement is weird because p(y|D, x, y∗) is not a truncated Gaussian distribution even when noiseless case. This is an approximation. In the noiseless case, the predictive distribution given the optimal value is at least should satisfy [C1] f(x) < y* for 'ALL' x and [C2] there exists x such that f(x) = f*. Note that here I assume the condition on x* is already omitted (e.g., the hessian condition) as I already mentioned. The predictive distribution under these condition is not a simple truncated Gaussian distribution anymore. More importantly, about this particular issue, the paper does not provide any novel treatment compared with existing studies. Although the authors seems claim novelty about the noise treatment, the noisy case also has been discussed in the past work (e.g., [15,22,39]).

- In line 142-143, the author described

> after conditioning on f∗, the posterior predictive density over y is a sum of a truncated Gaussian distribution over f and the Gaussian noise \epsilon

Related to the above comments, this description is also wrong and misleading. As already mentioned, the conditioning on f* is more complicated. It is not a simple sum of truncated Gaussian and the noise Gaussian, which is just an approximation (note that [26] explicitly declared they use an approximation which simplifies [C1] to a condition only on input query throughout the paper (though [C2] is implicitly omitted)).

- In line 183, the author described

> conditioning on f∗ yields a truncated normal distribution p(f|D, f∗), but the entropy is computed with regard to the density over noisy observations, y = f + \epsilon, which leads to an intractable entropy.

Under the replacement [C1, C2] to 'f(x) <= f* only for current input x' (the authors (implicitly) employs the almost same simplification), the computation of entropy of p(y | D, f*) is shown by [39] (Appendix C), in which f* is noiseless and y is additive noise contaminated. It is shown that the computation is tractable.

- Section 3.5 has no relation with the information theoretic approach. Further, I do not agree with selecting the argmax of the posterior mean mitigates the model misspecification issue. Using the posterior mean can be seen as fully exploiting the current model knowledge, and thus, it can be even risky if the model is misspecified. A rational that inverse-\gamma-greedy approach is not fully clear to me.

- Figure 1 is misleading and difficult to interpret. It misleads that JES identifies the correct maximizer of f(x), but at the moment, GP does not accurately predict the true function around the maximizer. As indicated by p(x*|D) in the figure, currently, around the maximizer of PES has the highest probability density having the maximizer. Therefore, the maximizer of a sample path from GP should have the largest value around the maximizer of PES with much higher probability around the maximizer of JES. In this sense, f(x) of Figure 1 (black line) shows a rare case which is convenient for the authors. Further, for me, the suggestions of next querying by all of the three acquisition functions in the plot are reasonable. It is difficult to judge which one is most appropriate from this figure.

- The information I((x,y),(x*,f*)) is obviously difficult to evaluate than PES and MES (because the dimension of the entropy is higher). On the other hand, the approximation employed by the proposed method is a naive moment matching, which would be a stronger approximation than the computational approaches taken in those past works (PES employs the expectation propagation, and MES evaluates the entropy of the truncated Gaussian without using moment matching (for which [39] shows noisy case as well)).

- Rationale for considering joint entropy of (x*,f*) is not fully clear to me. The purpose of a black-box optimization problem in general is to find x* = argmax f(x). Then, considering entropy p(x*) is quite natural. The Vapnik's principle says that you should not solve a problem more general than the problem of interest. In this sense, I consider that minimizing the entropy of p(x*,f*) can be more difficult than identifying x* only.

- The authors describe when JES is applied to a noiseless case, it has the infinite information gain at the sampled optimal point. I guess this phenomenon would be an artifact caused by approximations the authors (sometimes implicitly) introduced. I even think this phenomenon is intriguing and worth studying, rather than simply avoiding it by adding the noise term.

==================================================
 Post-rebuttal
==================================================

Thank you for providing many replies (and sorry that I could not reply during the feedback period). However, overall, I am not convinced by many of the responses.

<1> Novelty about noise treatment

Thank you for reflecting the comment. Since the authors explicitly cited the past works that deal with the observation noise in the original version, for me, it is quite surprising that the authors claimed novelty about the noise treatment, which is obviously not novel (To be honest, this degraded reliability of the paper).

<2> Misunderstanding about incorporation of f* globally

From the response, I consider that the authors have a fundamental misunderstanding about the discussion of the global truncation by f*.

I'd like to first note that the fact that p(f(x) | D, f*) is NOT a truncated normal distribution. This is an approximation. To confirm it, let me review the derivation of the predictive distribution of GPR. To simplify the discussion, I consider the noiseless setting (even in the noisy case, essential discussion is same).

In the usual (non-truncated) GP derivation, a predictive distribution p(f(x) | D) for a specific x is derived by (implicitly) marginalizing all other x' in the domain, which is possible because f(x) is jointly Gaussian for any x (sometimes called marginalization property).

On the other hand, in p(f(x) | D, f*), this marginalization becomes difficult, because the marginalization must be performed with a distribution in which f(x) is truncated for \forall x (given f*, f(x) \leq f* should hold for \forall x). In general, it is known that a marginal distribution of multivariate truncated Gaussian is not a truncated Gaussian (e.g., see Horrace 2005).

In this sense, representing p(f(x) | D, f*) as a truncated Gaussian is a local truncation-based approximation (in other words, marginalization is performed WITHOUT truncating other f(x') (x' \neq x)). Please note that what the authors claimed 'global' just means applying this local approximation for all candidate x when it is queried (i.e., it is not the global truncation in the sense described here).

The entire paper is seemingly written on the premise of this misunderstanding. For example, discussion in Sec3.2 implicitly assumes p(f(x) | D, f*) is a truncated Gaussian. Further, the discussion of '2. Dsregarding x* as optimumu' in the response, i.e., the Lipschitz based justification, is not correct (because except for the current queried x, f(x) is not truncated). Some other responses about truncated Gaussian would also be related.

Many information theoretic existing studies have employed this local approximations. For example, papers of Rectified MES [26] and PES [36] clearly explained about the local truncation, though the authors ignore it even after I pointed out in the first review. Similar to <1> above, here again, this makes me feel that the authors might not carefully read past works.

W. C. Horrace, Some results on the multivariate truncated normal distribution, Journal of Multivariate Analysis, 2005

<3> JES with a noiseless model

Although the author explains noiseless JES has the infinite information gain because p(y | D U (x*,f*),x,f*) is deterministic at x*, this rather suggests the difficulty of dealing with joint information gain of (x*,f*) appropriately. It makes the finite sample MC approximation of the expectation in (11) difficult, and obviously, just adding the observation noise is not an essential remedy. Further, the authors mentioned when L points exists (the number of optimal point samplings), noiseless JES becomes Thompson sampling (TS) (i.e., sampling from p(x*)) by using a simple heuristics. This also makes justification as a mutual information (MI) approximation unclear, because it seemingly indicates that the quantity cannot be interpreted as an MI approximation even with a large L.

<4> Existing studies for truncation by f* in the noisy setting (y is not truncated)

> Regarding the tractability of the computation: As the reviewer may agree on, we are after Eq. (24) in [39], Appendix C. Below Eq. (24), it is clearly stated that
> > The integral in (24) can be calculated by using numerical integration in the same way as (6).
>
> which indicates that the computation is not tractable, but needs approximation.

I'd have say that this reply is quite misleading. [39] intends that it is computationally tractable (though of-course it is analytically intractable). Their acquisition function actually performs the same type of numerical integration in the multi-fidelity setting, in which the highest fidelity is truncated but lower fidelity is not truncated (note that 1d numerical integration is often computationally easy), which is technically almost same as f(x) is truncated but y is not truncated also shown by the same paper. For me, this also seems that the authors might not pay sufficient attention to past works.

---

> ### Author Response · Authors · 2022-07-31
> **Response to Reviewer 7H23 (4/4)**
>
>
> ### 10. Intuition for Joint Entropy
>
> > ##### Rationale for considering joint entropy of $(x^*,f^*)$ is not fully clear to me (…) I consider that minimizing the entropy of $p(x^*,f^*)$ can be more difficult than identifying $x^*$ only.
>
>
> This statement has partly been addressed in part in (4). While we agree that the consideration of $p(x^*)$ is the most intuitive, there are two good reasons for the joint entropy:
>
> 1. The success of MES indicates that the information conveyed by $p(y^*)$ is valuable in solving the task at hand, so there is value in incorporating it.
> 2. Considering $p(x^*,f^*)$ as opposed to $p(x^*)$ yields drastically simplified calculations. We need only to treat $(x^*,f^*)$ as an observation in the GP and truncate the posterior distribution on $f^*$, as opposed to the numerous approximations in ES/PES.
>
> So, while finding $x^*$ _and_ $f^*$ may be more challenging than finding $x^*$, there is a tradeoff where the information that we gain regarding $f^*$ (for instance that the maximum is likely below some upper bound C) may help us in finding $x^*$ quicker (for a similar reason as the one that makes MES work well).
>
>
>
> ### 11. JES with a noiseless model
>
> > ##### The authors describe when JES is applied to a noiseless case, it has the infinite information gain at the sampled optimal point. I guess this phenomenon would be an artifact caused by approximations the authors (sometimes implicitly) introduced…
>
>
> The aforementioned artifact is a consequence of how non-random quantities are treated in a differential entropy framework. In the noiseless case, $p(y|\mathcal{D} \cup (x^*, f^*), x, f^*)$ is deterministic at $x^*$. As such, the entropy is negative infinity, and the information gain is infinite. If we were to compute the expected information gain using only one optimal sample $(x_\ell^*, f^*_\ell)$, our approach trivially reduces to Thompson Sampling.
>
> _So, what happens if we have more than one sample to compute the expectation?_
>
> In that case, we will have $L$ points (every sampled optimizer location) with infinite information gain. If we employ tie-breaking uniformly at random (which is typical in the BO literature) between all these points, we randomly choose 1 out of $L$ Thompson samples as the next query, which would once again be equivalent to standard Thompson Sampling.
>
> We hope that we have clarified the reviewer's concerns, and that you agree with our view on information-theoretical acquisition functions in prior work as well as the theoretic legitimacy of our proposed approach. If this is the case, we would greatly appreciate it if you would consider increasing your score.
>
>
>
> ### Other questions:
> > ##### In the experiment, what does 'end of the initial design' mean? Why is performance of each method is different at that point? (e.g., in Fig 5 and 7)
>
> It is the initialization phase, where queries are selected randomly. The dashed line is the first iteration where queries are selected according to the acquisition function, and not randomly. The initial design locations are per-seed identical across acquisition function. In Fig. 5, we plot inference regret. While the initialization is identical (and the inferred optima) we observe that PES, after the first iteration, infers that the optimum is in a region where it evidently is not (on average), which explains the upwards spike for PES specifically.
>
> In Fig. 7, on Hartmann-3, we have actually made a mistake, where new MES runs were appended to old, crashed ones. This has now been corrected, and the Hartmann-3 plot has been updates. MES performs better, particularily in earlier iterations. Thanks a lot for catching this error.
>
> ## References:
> HB Moss, DS Leslie, J Gonzalez, P Rayson. GIBBON: General-purpose Information-Based Bayesian OptimisatioN. _Journal of Machine Learning Research_ (JMLR). 2021.\
> Shion Takeno, Tomoyuki Tamura, Kazuki Shitara, Masayuki Karasuyama. Sequential and Parallel Constrained Max-value Entropy Search via Information Lower Bound. _Proceedings of the 39th International Conference on Machine Learning_. 2022.

---

> ### Author Response · Authors · 2022-07-31
> **Response to Reviewer 7H23 (3/4)**
>
> ### 7. Truncated Gaussian statement
>
> > ##### About MES, in line 103-104, the authors described
> $p(y|\mathcal{D}, x, y^*)$ _is a truncated Gaussian distribution, for which the entropy can be computed in closed form. However, $p(y|\mathcal{D}, x, y^*)$ takes this form only in a strictly noiseless setting_
>
> > ##### This statement is weird because $p(y|\mathcal{D}, x, y^*)$ is not a truncated Gaussian distribution even when noiseless case. This is an approximation. In the noiseless case, the predictive distribution given the optimal value is at least should satisfy [C1] $f(x) \leq y^*$ for 'ALL' $x$ and [C2] there exists $x$ such that $f(x) = f^*$...
>
>
> The novelty of the noise treatment has been addressed in (1).
>
> If the reviewer is referring to the fact that C2 may not be satisfied in MES or any if the aforementioned follow-ups, we agree — there is no mechanism in MES that ensures that the optimal value $f^*$ is actually obtained by any point in the search space, only that all values are _lower_ than said maximum. While asserting the correctness of C2 for other acquisition functions is outside the scope of our work, we note that JES directly satisfies C2 by conditioning on $(x^*, f^*)$ and C1 by truncating predictions for $f(x)$ at $f^*$ for ALL $x$. (We believe [C1] should be $f(x) \leq f^*$, not $f(x) \leq y^*$ as stated, which we discuss in (5, 6)), The conditioning on $x^*$ is not omitted, but naturally follows from our conditioning, as covered in (3). We added the note regarding C2 in the updated version of the paper.
>
>
>
> ### 8. Truncated Gaussian statement, part 2
>
> > ##### In line 183, the author described
> _conditioning on $f^*$ yields a truncated normal distribution $p(f|\mathcal{D}, f^*)$, but the entropy is computed with regard to the density over noisy observations, $y = f + \epsilon$, which leads to an intractable entropy._
>
> > ##### Under the replacement [C1, C2] to '$f(x) \leq f^*$ only for current input $x$ (the authors (implicitly) employs the almost same simplification), the computation of entropy of $p(y |\mathcal{D}, f^*)$ is shown by [39] (Appendix C), in which $f^*$ is noiseless and $y$ is additive noise contaminated. It is shown that the computation is tractable.
>
> The admittedly dubious notation $p(f|\mathcal{D}, f^*)$ has been addressed in (2), where it is also explained that it is in fact the distribution $p(y|\mathcal{D}\cup (x^*,f^*), x, f^*)$ that is considered. Nevertheless, this distribution also requires truncation wrt. $f$.
>
> Like the aforementioned previous works [26, 29, 39] (old version reference indexing), Moss et. al. (2021) and Takeno et. al. (2022), we condition on an optimal noiseless value $f^*$ globally by truncating the posterior over $f$. Since we still have noisy outputs, the posterior over $y$ is not a truncated normal. There is no local or approximating aspect to this conditioning. This is covered in greater detail in (5). C1 and C2 are adressed in (7).
>
> Regarding the tractability of the computation: As the reviewer may agree on, we are after Eq. (24) in [39], Appendix C. Below Eq. (24), it is clearly stated that
>
> _The integral in (24) can be calculated by using numerical integration in the same way as (6)._
>
> which indicates that the computation is **not** tractable, but needs approximation.
>
>
>
> ### 9. Inverse Gamma-greedy queries
>
> > ##### Section 3.5 has no relation with the information theoretic approach. Further, I do not agree with selecting the argmax of the posterior mean mitigates the model misspecification issue…
>
>
> Selecting the argmax of the posterior mean can indeed be seen as fully exploiting the current model knowledge — or at least the _presumed_ knowledge. As such, it is also the simplest way to _disprove_ a model, if the presumed optimum is incorrect. Information-based acquisition functions carry a belief over the location of the optimum, but do not have an inherent incentive to validate this belief. The rationale of the inverse $\gamma$-greediness is that it is a mechanism that validates the aforementioned belief over the location of the optimum. If the believed optimum is drastically incorrect, we can establish some level of misspecification and allow JES (or PES/MES) to recalibrate its belief. As such, the modeled density over the optimum (which likely corresponds well with this argmax) will shift substantially, if the model is indeed misspecified. Notably, this cannot be done by choosing explorative queries because there is no knowledge to check in that case.
>
> We are surprised by the comment that evaluations at the argmax of the posterior mean are “risky” if the model is misspecified. The only risk is that the model is incorrect, which is actually not inherently bad. Rather, it allows the model to fit better afterwards; as such, it guards against the risk that that point is not actually as good as expected by the model.
>
> In Appendix C.2 (Previously Appendix F), we show that the inverse $\gamma$-greediness helps PES and MES as well.

---

> ### Author Response · Authors · 2022-07-31
> **Response to Reviewer 7H23 (2/4)**
>
> ### 4. Difficulty of evaluation
>
> > ##### The information $I((x,y),(x^*,f^*))$ is obviously more difficult to evaluate than PES and MES (because the dimension of the entropy is higher). On the other hand, the approximation employed by the proposed method is a naive moment matching…
>
>
> We indeed consider a higher-dimensional density than that in PES and MES, and we agree that this could potentially indicate that the evaluation of the acquisition function would be more difficult. However, the dimensionality of the problem alone does not determine the dfficulty of evaluation. Rather, it is determined by (a.) the quality of the employed approximation and (b.) the modality of the density which MC is performed over (which admittedly scales with dimensionality).
>
> (a.) THe approximation in JES is a minor part of the total information gain. As demonstrated in Fig. 3, the majority of the information gain comes from the first, exact, step of JES, which conditions the GP on the observation $(x^*, f^*)$; this makes the moment matching approximation less significant. In Appendix E, Figure 13, the quality of the moment matching approximation compared to an asymptotically exact MC approximation of the entropy for various levels of noise and truncation.
>
> (b.) It is demonstrated in Appendix D that JES is robust with regard to the number of MC samples (20, 100 and 500 samples show almost identical performance). The joint density will typically cluster in a few attractive regions of the search space, which is demonstrated in Fig. 3. Notably, PES and JES have the same number of modes in this case, but those of PES are admittedly less concentrated. Approximation will likely be difficult initially (when all acquisition functions are relatively uninformed), but naturally improves over time.
>
>
> ### 5. Incorporation of $f^*$ globally
>
> > ##### The proposed method incorporates ($x^*$,$f^*$) into GP to calculate $p(y|\mathcal{D},f^*)$, but $f(x) \leq f^*$ is only (approximately) imposed for the current queried x. (...) Similar approximations have been employed by past works, but in the paper, this approximation is implicitly introduced…
>
>
> We believe there might be a misunderstanding here. Like MES, we truncate the posterior (on f, i.e., $p(f|\mathcal{D}, f^*)$), which is naturally done globally, and not, as the reviewer states, only locally for the current queried x. The precise operation should be evident from Fig. 2 (right), also showing the global nature of the conditioning visually. Other work [26, 39, 43] support that this operation is global. Like [26, 39] and Moss et. al. (2021), we make the important distinction that the posterior on f is truncated, whereas the posterior on y is not. Consequently, all the aforementioned works employ approximations of various kinds to approximate the entropy over the distribution over the noisy outputs $y$, but there is no approximation required for the conditioning on $f^*$. The important noisy output distinction is missing in the original MES paper [43].
>
> If the reviewer has a different viewpoint on this, we encourage further clarification.
>
>
>
> ### 6. $f^*$ / $y^*$ differentiation
>
> > ##### About MES, the authors described that MES considers $y^* = f(x^*) + \epsilon$ (according to the descriptions after (6)). However, this is not the case. In the original MES paper [43], $y^*$ is defined as $f(x^*)$ (noise is not included in $y^*$). Therefore, although the authors claim the replacement $y^*$ with $f^*$ is one of differences from MES after (9), it is also misleading.
>
>
> At lines 39-40 we state that:
>
> _…it (MES) does not differentiate between the noiseless optimal value $f^*$ and some noisy maximal value, $y_{max}$._
>
> Moreover, at lines 104-105 we state that
>
> $p(y|\mathcal{D}, x, y^*)$ _takes this form only in a strictly noiseless setting [26 , 39], where it holds true that $f^* = y^*$ (should be $y_{max}$, updated in new version)._
>
> As mentioned in (5), and as evidenced by the references [26 , 39] as well as Moss et. al. (2021, section 2.1), we are not the first to state that the noiseless optimal value and a potential noisy maximum are identical only in a noiseless setting, and that the definition in MES does not align with their proposed approach. It should be intuitively clear that $y_{max} \neq f(x^*)$ in general, which is why the definition in MES is not consistent, and consequently, that the posterior distribution over $y$ when conditioned on the optimal value is not a truncated normal in general.

---

> ### Author Response · Authors · 2022-07-31
> **Response to Reviewer 7H23 (1/4)**
>
> We thank the reviewer for the thorough critique of our paper. We have addressed some of your concerns and incorporated them into an updated version of our paper. However, we politely disagree with the claims regarding the theoretical shortcomings of our approach. We hope our responses will bring this to light.
>
>
> ### Initial Statement
> We believe there is a fundamental disagreement regarding notation and meaning of $f^*, y^*, f$ and $y$. We outline our notation regarding below, which differs from original MES [43], but is consistent with MES follow-up works [26, 29, 39] (old version reference indexing), Moss et. al. (2021) and Takeno et. al. (2022):
>
> $f^*$ is, by definition, the _noiseless_ optimal value. $y(x)$ is the noise-contaminated observation of a true function value $f(x)$. as such, $y(x)$ does not have to satisfy $y(x) \leq f^*, \;  \forall x$ when we condition on an optimal value $f^*$. However, $f(x) \leq f^*, \;  \forall x$, must hold. This concept is displayed in Fig. 2 (right), where noise can cause observations to surpass the true optimal value $f^*$. Moreover, the notion of a $y^*$, i.e., a noisy optimal value, is meaningless. This is best illustrated if we were to query at $x^*$ multiple times in a row, as we would get different noisy $y^*$-values with every query.
>
>
> ### 1. Noise claim
>
> > ##### Although the authors implicitly assumes conventional MES is only for noiseless, it is not the case. [22] and [39] have already shown how to incorporate the additive noise. (…) Therefore, the description of the third main advantage listed in the end of introduction is misleading…
>
>
> We agree with the reviewer that the paper was misleading wrt. where [26, 39] stand on observation noise (we believe that the reviewer was referring to [26] instead of [22]?); we have now expanded our description of those contributions, and we are more explicit in the introduction and related work sections. We have also rephrased the third advantage in our contributions, removing the noise aspect.
>
>
> ### 2. Disregarding x* as optimum
> > ##### From (8) to (9), $p(y|\mathcal{D},x,x^*,f^*)$ is re-written as $p(y|\mathcal{D} \cup (x^*,f^*),x,f^*)$. After that, in Section 3.4 (which describes the computations of the entropy of $p(y|\mathcal{D}\cup (x^*,f^*),x,f^*)$, it is further re-written as p$(y|\mathcal{D},f^*)$. This re-writing hides the fact that the original $p(y|\mathcal{D}\cup x,x^*,f^*)$ contains a condition that $x^*$ is the maximizer'...
>
> While admittedly ambiguous in the paper, $p(y|\mathcal{D} \cup (x^*,f^*),x,f^*)$ is not rewritten as p$(y|\mathcal{D},f^*)$, as evidenced by Eq. (12) and Eq. (13) in the new pdf (11 & 12 in the old version). p$(y|\mathcal{D},f^*)$ is used strictly to explain what happens to the posterior over $y$ when conditioning on $f^*$. We have removed this ambiguity in the new version of the paper.
>
> Regarding convexity (or in a maximization case, concavity): By conditioning on the observation $(x^*,f^*)$, and on $f^*)$ being the optimal value, we effectively say that we have an observation in $x^*$ that takes on the value $f^*$, and that after truncation no other noiseless function value in the search space can be larger. As such, we know that we have concavity at $x^*$, since f must be non-increasing around $x^*$. More percisely, we have that:
>
> - $f$ is Lipschitz-continous
> - $f(x^*) = f^*$, $f(x) \leq f^*, \forall x$
> - $\frac{1}{2}(f(x^* + \frac{1}{2}\vec{r}) + f(x^* - \frac{1}{2}\vec{r})) \leq f^* = f(x^*)$ for some sufficiently small but positive $\rho$, s.t. $|\vec{r}| \leq \rho$,
>
> which means that we have concavity in a $\rho$-ball around $x^*$, so the function is indeed concave at the conditioned optimum.
>
> In Fig. 2 (right), we display the GP that models $p(y|\mathcal{D} \cup (x^*,f^*),x,f^*)$. We hope that it is evident from the density in yellow that our conditioning makes $x^*$ the optimum, as no point in the search space has a positive density on larger (noiseless) values than $f^*$.
>
>
>
> ### 3. Misleading figure
>
> > ##### Figure 1 is misleading and difficult to interpret. It misleads that JES identifies the correct maximizer of $f(x)$, but at the moment, GP does not accurately predict the true function around the maximizer. As indicated by $p(x^*|\mathcal{D})$ in the figure, currently, around the maximizer of PES has the highest probability density having the maximizer…
>
>
> Figure 1 is a toy example intended to show the emphasis of each acquisition function. We don't claim that the PES and MES proposed queries are wrong and we agree with the reviewer that the three proposed query points are all reasonable. The essence of Figure 1 is that JES puts more emphasis in learning about both $x^*$ and $f^*$. The choice of black-box function (black) and observations (red) is driven by the JES general behavior that we want to highlight; this behavior is visible in other examples as well.

---

> ### Author Response · Authors · 2022-08-06
> **Additional response to Reviewer 7H23**
>
> We once again thank the reviewer for the initial feedback. We would love to hear from the reviewer if our previous answers addressed the reviewer's concerns, and to engage in a discussion about any misunderstandings we may have.
>
> As authors, we are only allowed to respond until Tuesday evening. As such, we would appreciate a response, as it would facilitate a continued discussion of our paper and how it relates to other information-theoretic acquisition functions. Thank you for your time!

---

### Official Review · Reviewer_Lu7F · 2022-07-22

**Rating:** 6
**Confidence:** 3
**Soundness:** 3 good
**Presentation:** 3 good
**Contribution:** 2 fair

**Summary:**

In this paper, the authors propose a Bayesian optimization method for black-box optimization problems using a new entropy search-type acquisition function that combines the advantages of predictive entropy search and max-value entropy search. The proposed method, joint entropy search (JES),  uses the mutual information between the optimal solution/optimal value pair (x^*, f^*) and the candidate query points (x, y) as the acquisition function. JES, like other entropy search acquisition functions, is expressed as the difference of two differential entropy terms. The main difference from other entropy search is that the second term takes the expected value of entropy with respect to the optimal pair (x^*, f^*).

To compute this expectation with MC sampling, a parametric model representation of the kernel function is obtained using Bochner's theorem. By sampling its parameters, a sample path from the GP is obtained in the form of a basis function model, whose maximum solution and maximum value are estimates of x^* and f^*, respectively. In particular, the distribution of f conditioned on the maximum f^* is a truncated normal distribution and is approximated by moment matching. Like other entropy search type acquisition functions, JES can be greatly affected by model misspecification. Hence, in the proposed algorithm, the inverse gamma greedy method is used to ensure that the JES search and the search using the predictive mean as the acquisition function are executed with a certain probability each, so that the effect of model misspecification does not remain until later.

In the experiments, the authors first explained the difference in behavior between JES and PES or MES through optimization of the benchmark function, and showed experimental results suggesting that JES is effective. The practical performance of JES compared to PES and MES was also evaluated by the MLP hyperparameter optimization task.

**Questions:**

- (weakness) Interpretation of the inverse gamma greedy method is difficult. In terms of mitigating the effects of model misspecification, it seems to me that acquisition by predictive variance that represents "exploration" is better than acquisition by predictive mean that represents "exploitation". Could you explain in detail that why does the latter mitigate the problem?

- Is the \gamma of the inverse \gamma greedy method a fixed value? Or does it require appropriate scheduling like \varepsilon in the \varepsilon-greedy method?

- (weakness) In the real task (MLP hyperparameter optimization), I am concerned that the performance of JES (that is considered robust to noisy observations) is not so different from that of MES in terms of accuracy relative to the number of observation points. I think that the robustness of JES should be verified in more detail with artificial data that controls the amount (and/or structures) of noise.

**Limitations:**

The authors adequately addressed the limitations and potential negative societal impact of their work.

**Strengths And Weaknesses:**

Strength

- In the entropy search acquisition function, using the optimal solution x^* and the optimal value f^* simultaneously as conditions is expected to increase the "amount of information" available for the search compared to PES and MES, which consider only one of each.

- The inverse gamma greedy method is proposed to mitigate the influence of model misspecification, which is a property of all entropy search type acquisition functions including PES and MES. Although this idea is a heuristic, it is considered to be a general-purpose method that can be applied to other entropy-search type acquisition functions.

- Experiments suggest that JES is more robust to observation noise than PES or MES. This has important implications for problems involving measurement errors of various magnitudes, such as medical research or materials science.

Weakness

see Questions

---

> ### Author Response · Authors · 2022-07-31
> **Response to Reviewer Lu7F**
>
> We thank the reviewer for the thorough feedback. Your questions are addressed in our responses below.
>
>
> ### 1. Inverse-gamma interpretation & value
>
> > ##### Interpretation of the inverse gamma greedy method is difficult. (…) Could you explain in detail that why does the latter mitigate the problem?
>
> Selecting the argmax of the posterior mean can indeed be seen as fully exploiting the current model knowledge — or at least the _presumed_ knowledge. As such, it is also the simplest way to _disprove_ a model, if the presumed optimum is incorrect. Information-based acquisition functions carry a belief over the location of the optimum, but do not have an inherent incentive to double-check this belief. The inverse $\gamma$-greediness acts as a mechanism that checks the aforementioned belief about the location of the optimum. If the believed optimum (i.e.) the argmax of the mean function) is drastically incorrect, we can establish some level of misspecification and allow JES (or PES/MES) to recalibrate its belief. As such, the modeled density over the optimum (which likely corresponds well to this argmax) will shift substantially, if the model is indeed misspecified. Notably, this cannot be done by choosing explorative queries because there is no knowledge to check in that case.
>
> We employ a fixed rate of inverse-greedy queries, which is 10% in our experiments. In Appendix C.1, an ablation study on this rate is performed, but the range 5%-20% yields very similar results for all synthetic test functions. Varying rates have not been tested.
>
> In Appendix C.2 (Previously Appendix F), we show that the inverse $\gamma$-greediness helps PES and MES as well.
>
>
> ### 2. MLP optimization
>
> > ##### In the real task (MLP hyperparameter optimization), I am concerned that the performance of JES (that is considered robust to noisy observations) is not so different from that of MES…
>
> In Appendix C.2, we display the performance of JES, PES and MES in a synthetic setting with various noise levels, ranging from very low (the BOTorch minimum permitted variance) to very high ($\sigma_\epsilon$ = ~25% of the output range) and show that JES substantially outperforms MES and PES. In the synthetic setting, we can measure the true, noiseless observations (and add synthetic noise on top), whereas we do not for the MLPs, and have to make due with measuring the performance as measured on the noisy observations. This naturally causes noise in the measurements for the MLP. If the reviewer has other experiments in mind to alleviate this concern, we would be very happy to hear about them and carry them out.
>
> We again thank the reviewer for the helpful review and we hope that our answers helped clarify your concerns. We would appreciate it if the reviewer would consider increasing the score if that is the case.

---

> ### Author Response · Authors · 2022-08-07
> **Additional response to Reviewer Lu7F**
>
> We once again thank the reviewer for the initial feedback.
>
> > ####  I am concerned that the performance of JES (that is considered robust to noisy observations) is not so different from that of MES in terms of accuracy relative to the number of observation points..
>
> We have now added results from three additional MLP tasks from HPOBench in Appendix G. Given the rebuttal time constraint, we prioritized the most competitive methods JES, MES and EI, and ran only 45 repetitions; we will add PES and the remaining repetitions in the camera ready. In short, JES outperforms MES on 4/6 tasks, and is always at least as good as MES.
>
> The 3 new benchmarks show that JES outperforms the baselines in one out of three benchmarks (German Credit). In Image Segmentation and Software Defects, JES outperforms MES but not EI (EI and JES are mostly on-par on Software Defects). Aggregating the results on the 6 HPOBench real-world benchmarks, we see that JES outperforms all the baselines on 2/6 of the benchmarks (Blood Transfusion and German Credit) and is only outperformed once. While it is outside of the scope of this work to extensively compare JES with non information-theoretical acquisition functions, we notice that EI only outperforms JES 1/6 (Image Segmentation) and is either on par 3/6 (Software Defects, Vehicle Silhouette, and Credit Application) or worse 2/6 than JES (German Credit and Blood Transfusion).
>
> Along with the GP sample, noisy GP sample and synthetic function experiments previously presented in the paper (on which JES is consistently superior), we hope that this convinces the reviewer that JES is not only a relevant new acquisition function with promise for future work, but that it also leads to SOTA performance among information-theoretic acquisition functions. We are happy to consider any additional feedback the reviewer may have and discuss further.

---

> > ### Comment · Reviewer_Lu7F · 2022-08-08
> > **Thank you for the careful responses**
> >
> > Thank you for your careful responses to my concerns.
> > The comments I received helped me understand the inverse gamma greedy method.
> > Based on the responses I have received and the comments of other reviewers, I would like to raise my score by one.

---

### Meta-Review · Area_Chair_mHCZ · 2022-08-24

**Recommendation:** Accept
**Confidence:** Certain

**Metareview:**

The authors propose a Bayesian optimization method for black-box optimization problems using a new entropy search-type acquisition function that combines the advantages of predictive entropy search and max-value entropy search. The proposed method, joint entropy search (JES), uses the mutual information between the optimal solution/optimal value pair (x^*, f^*) and the candidate query points (x, y) as the acquisition function. In the experiments, the authors first explained the difference in behavior between JES and the baselines PES or MES through optimization of the benchmark function, and showed experimental results suggesting that JES is effective. The practical performance of JES compared to PES and MES was also evaluated by the MLP hyperparameter optimization task.

Strengths:

1 - The intuition behind the basic idea is simple.
2 - A variety of empirical evaluations are provided.
3 - The inverse gamma greedy method is proposed to mitigate the influence of model misspecification, which is a property of all entropy search type acquisition functions including PES and MES.
4 - Experiments suggest that JES is more robust to observation noise than PES or MES.

Weaknesses:

1 - While the method is very interesting, it also feels like an incremental step from PES and MES.
2 - As pointed out by reviewer 7H23, the authors incorrectly state that conditioning on f* yields a truncated Gaussian predictive distribution. Reviewer 7H23 correctly states that "it is known that a marginal distribution of multivariate truncated Gaussian is not a truncated Gaussian". The authors should revise the paper to indicate that they make the additional approximation that conditioning on f* yields a truncated Gaussian predictive distribution.

Decision:

A majority of reviewers are strongly positive about the paper. The only negative reviewer, 7H23, points out minor limitations, with the exception of the mistake made by the authors, as described in point 2 in the list of weaknesses above. The authors need to update the paper to clarify that they make the additional approximation that conditioning on f* yields a truncated Gaussian predictive distribution.

**Award:**

No

---

### Decision · Program_Chairs · 2022-09-14

Accept